# Learning Optimal Contracts: How to Exploit Small Action Spaces

**Francesco Bacchiocchi, Matteo Castiglioni, Alberto Marchesi & Nicola Gatti**
Politecnico di Milano, Milan, Italy
`{name.surname}@polimi.it`

## Abstract

We study *principal-agent problems* in which a principal commits to an outcome-dependent payment scheme—called *contract*—in order to induce an agent to take a costly, unobservable action leading to favorable outcomes. We consider a generalization of the classical (single-round) version of the problem in which the principal interacts with the agent by committing to contracts over multiple rounds. The principal has no information about the agent, and they have to learn an optimal contract by only observing the outcome realized at each round. We focus on settings in which the *size of the agent's action space is small*. We design an algorithm that learns an approximately-optimal contract with high probability in a number of rounds polynomial in the size of the outcome space, when the number of actions is constant. Our algorithm solves an open problem by Zhu et al. (2023). Moreover, it can also be employed to provide a $\widetilde{\mathcal{O}}(T^{4/5})$ regret bound in the related online learning setting in which the principal aims at maximizing their cumulative utility over rounds, considerably improving previously-known regret bounds.

## 1 Introduction

The computational aspects of *principal-agent problems* have recently received a growing attention in algorithmic game theory (Dütting et al., 2019; Alon et al., 2021; Guruganesh et al., 2021; Dutting et al., 2021; Dütting et al., 2022; Castiglioni et al., 2022; Alon et al., 2023; Castiglioni et al., 2023a;b; Dütting et al., 2023; Guruganesh et al., 2023; Deo-Campo Vuong et al., 2024; Dutting et al., 2024). Such problems model the interaction between a principal and an agent, where the latter takes a costly action inducing some externalities on the former. In *hidden-action* models, the principal cannot observe the agent's action, but only an outcome that is stochastically obtained as an effect of such an action and determines a reward for the principal. The goal of the principal is to incentivize the agent to take an action leading to favorable outcomes. This is accomplished by committing to a *contract*, which is a payment scheme defining a payment from the principal to the agent for every outcome.

Nowadays, thanks to the flourishing of digital economies, principal-agent problems find applications in a terrific number of real-world scenarios, such as, *e.g.*, crowdsourcing platforms (Ho et al., 2014), blockchain-based smart contracts (Cong & He, 2019), and healthcare (Bastani et al., 2016). Most of the works on principal-agent problems focus on the classical model in which the principal knows everything about the agent, including costs and probability distributions over outcomes associated with agent's actions. Surprisingly, the study of settings in which the principal does *not* know all agent's features and has to *learn* them from experience has been neglected by almost all the previous works, with the exception of (Ho et al., 2014; Zhu et al., 2023; Cohen et al., 2022; Han et al., 2023).

In this paper, we consider a generalization of the classical hidden-action principal-agent problem in which the principal repeatedly interacts with the agent over *multiple rounds*. At each round, the principal commits to a contract, the agent plays an action, and this results in an outcome that is observed by the principal. The principal has no prior information about the agent's actions. Thus, they have to *learn an optimal contract* by only observing the outcome realized at each round. Our goal is to design algorithms which prescribe the principal a contract to commit to at each round, in order to learn an "approximately-optimal" contract with high probability by using the minimum possible number of rounds. This can be seen as the problem of establishing the *sample complexity* of learning optimal contracts in hidden-action principal-agent settings.

Ho et al. (2014) were the first to consider the problem of learning optimal contracts, though they focus on instances with a very specific structure, as the more recent work by Cohen et al. (2022). Recently, Zhu et al. (2023) addressed the problem of learning optimal contracts in general principal-agent settings. They provide an algorithm whose cumulative regret with respect to playing an optimal contract is upper bounded by $\widetilde{\mathcal{O}}(\sqrt{m} \cdot T^{1-1/(2m+1)})$, with an (almost-matching) lower bound of $\Omega(T^{1-1/(m+2)})$, where $m$ is the number of outcomes and $T$ the number of rounds. The result of Zhu et al. (2023) is very unpleasant when $m$ is large, since in such a case the regret grows almost linearly in $T$. Moreover, in the instances used in their lower bound, the number of agent's actions is exponential in $m$, leaving as an open problem establishing whether better guarantees can be obtained when the action space is small (see also the discussion in (Zhu et al., 2023)).

**Original contributions** We provide an algorithm that finds an "approximately-optimal" contract with high probability, requiring a number of rounds that grows polynomially in the size of the problem instance (including the number of outcomes $m$) when the number of agent's actions is a constant. The algorithm can be easily exploited to achieve cumulative regret (with respect to an optimal contract) upper bounded by $\widetilde{\mathcal{O}}(m^n \cdot T^{4/5})$, which polynomially depends on the instance size when the number of agent's actions $n$ is constant. This solves the open problem recently stated by Zhu et al. (2023). Our algorithm works by "approximately identifying" a covering of contracts into best-response regions, each one representing a set of contracts in which a given agent's action is a best response. The algorithm does so by progressively refining upper and lower bounds for such regions, until they coincide. One of the main challenges faced by our algorithm is that, after each round, the principal only observes an outcome stochastically determined by the agent's best response, rather than observing the action itself. This makes impossible identifying the action played by the agent. Our algorithm overcomes such an issue by working with *meta-actions*, which group together agent's actions associated with "similar" distributions over outcomes. Such meta-actions need to be "discovered" online by the algorithm, which thus has to update the set of meta-actions on the basis of the observed feedbacks. As a result, our algorithm has to adopt a trial-and-error approach which re-starts the covering procedure each time a new agent's meta-action is discovered.

**Relation to repeated Stackelberg games** Our work is related to the problem of learning an optimal commitment in repeated Stackelberg games (Letchford et al., 2009; Peng et al., 2019). However, differently from Stackelberg games, in principal-agent problems the principal cannot observe the action undertaken by the agent. This prevents us from relying on existing techniques (see, *e.g.*, Peng et al. (2019)) which identify the hyperplanes defining the best-response regions of the follower. As a consequence, we have to work with meta-actions that only "approximate" the set of actions and we can only compute "approximate" separating hyperplanes. Such approximations are made effective by the particular structure of principal-agent problems, in which an approximately incentive compatible contract can be turned into an incentive compatible one by only suffering a small utility loss (see, *e.g.*, (Dutting et al., 2021; Zhu et al., 2023)). This is *not* the case for Stackelberg games.

## 2 PRELIMINARIES ON HIDDEN-ACTION PRINCIPAL-AGENT PROBLEMS

An instance of the *hidden-action principal-agent problem* is characterized by a tuple $(\mathcal{A}, \Omega)$, where $\mathcal{A}$ is a finite set of $n := |\mathcal{A}|$ actions available to the agent, while $\Omega$ is a finite set of $m := |\Omega|$ possible outcomes. Each agent's action $a \in \mathcal{A}$ determines a probability distribution $F_a \in \Delta_\Omega$ over outcomes, and it results in a cost $c_a \in [0, 1]$ for the agent.[1] We denote by $F_{a,\omega}$ the probability with which action $a$ results in outcome $\omega \in \Omega$, as prescribed by $F_a$. Thus, it must be the case that $\sum_{\omega \in \Omega} F_{a,\omega} = 1$ for all $a \in \mathcal{A}$. Each outcome $\omega \in \Omega$ is characterized by a reward $r_\omega \in [0, 1]$ for the principal. Thus, when the agent selects action $a \in \mathcal{A}$, the principal's expected reward is $\sum_{\omega \in \Omega} F_{a,\omega} r_\omega$.

The principal commits to an outcome-dependent payment scheme with the goal of steering the agent towards desirable actions. Such a payment scheme is called *contract* and it is encoded by a vector $p \in \mathbb{R}_+^m$ defining a payment $p_\omega \geq 0$ from the principal to the agent for each outcome $\omega \in \Omega$.[2] Given a contract $p \in \mathbb{R}_+^m$, the agent plays a *best-response* action that is: (i) *incentive compatible* (IC), *i.e.*, it

---

[1] Given a finite set $X$, we denote by $\Delta_X$ the set of all the probability distributions over the elements of $X$.

[2] As it is customary in contract theory (Carroll, 2015), in this work we assume that the agent has *limited liability*, meaning that the payments can only be from the principal to the agent, and *not viceversa*.

maximizes their expected utility; and (ii) *individually rational* (IR), *i.e.*, it has non-negative expected utility. We assume w.l.o.g. that there always exists an action $a \in \mathcal{A}$ with $c_a = 0$. Such an assumption ensures that there is an action providing the agent with positive utility. This guarantees that any IC action is also IR and allows us to focus w.l.o.g. on IC only.

Whenever the principal commits to $p \in \mathbb{R}_+^m$, the agent's expected utility by playing an action $a \in \mathcal{A}$ is equal to $\sum_{\omega \in \Omega} F_{a,\omega} p_\omega - c_a$, where the first term is the expected payment from the principal to the agent when selecting action $a$. Then, the set $A(p) \subseteq \mathcal{A}$ of agent's best-response actions in a given contract $p \in \mathbb{R}_+^m$ is formally defined as follows: $A(p) := \arg\max_{a \in \mathcal{A}} \sum_{\omega \in \Omega} F_{a,\omega} p_\omega - c_a$. Given an action $a \in \mathcal{A}$, we denote with $\mathcal{P}_a \subseteq \mathbb{R}_+^m$ the *best-response set* of action $a$, which is the set of all the contracts that induce action $a$ as agent's best response. Formally, $\mathcal{P}_a := \{p \in \mathbb{R}_+^m \mid a \in A(p)\}$.

As it is customary in the literature (see, *e.g.*, (Dütting et al., 2019)), we assume that the agent breaks ties in favor of the principal when having multiple best responses available. We let $a^\star(p) \in A(p)$ be the action played by the agent in a given $p \in \mathbb{R}_+^m$, which is an action $a \in A(p)$ maximizing the principal's expected utility $\sum_{\omega \in \Omega} F_{a,\omega} (r_\omega - p_\omega)$. For ease of notation, we introduce the function $u : \mathbb{R}_+^m \to \mathbb{R}$ to encode the principal's expected utility under all the possible contracts; formally, the function $u$ is defined so that $u(p) = \sum_{\omega \in \Omega} F_{a^\star(p),\omega} (r_\omega - p_\omega)$ for every $p \in \mathbb{R}_+^m$.

In the classical (single-round) hidden-action principal-agent problem, the goal of the principal is find a contract $p \in \mathbb{R}_+^m$ that maximizes the expected utility $u(p)$. By letting $\mathsf{OPT} := \max_{p \in \mathbb{R}_+^m} u(p)$, we say that a contract $p \in \mathbb{R}_+^m$ is *optimal* if $u(p) = \mathsf{OPT}$. Moreover, given an additive approximation error $\rho > 0$, we say that $p$ is *$\rho$-optimal* whenever $u(p) \geq \mathsf{OPT} - \rho$.

## 3 LEARNING OPTIMAL CONTRACTS

We study settings in which the principal and the agent interact over multiple rounds, with each round involving a repetition of the same instance of hidden-action principal-agent problem. The principal has no knowledge about the agent, and their goal is to learn an optimal contract. At each round, the principal-agent interaction goes as follows: (i) The principal commits to a contract $p \in \mathbb{R}_+^m$. (ii) The agent plays $a^\star(p)$, which is *not* observed by the principal. (iii) The principal observes the outcome $\omega \sim F_{a^\star(p)}$ realized by the agent's action. The principal only knows the set $\Omega$ of possible outcomes and their associated rewards $r_\omega$, while they do *not* know anything about agent's actions $\mathcal{A}$, their associated distributions $F_a$, and their costs $c_a$.

The goal is to design algorithms for the principal that prescribe how to select a contract at each round in order to learn an optimal contract. Ideally, we would like an algorithm that, given an additive approximation error $\rho > 0$ and a probability $\delta \in (0, 1)$, is guaranteed to identify a $\rho$-optimal contract with probability at least $1 - \delta$ by using the minimum possible number of rounds.[3] The number of rounds required by such an algorithm can be seen as the *sample complexity* of learning optimal contracts in hidden-action principal-agent problems (see (Zhu et al., 2023)).

The following theorem shows that the goal defined above is too demanding.

**Theorem 1.** *For any number of rounds $N \in \mathbb{N}$, there is no algorithm that is guaranteed to find a $\kappa$-optimal contract with probability greater than or equal to $1 - \delta$ by using less than $N$ rounds, where $\kappa, \delta > 0$ are some suitable absolute constants.*

Theorem 1 follows by observing that there exist instances in which an approximately-optimal contract may prescribe a large payment on some outcome. Thus, for any algorithm and number of rounds $N$, the payments used up to round $N$ may *not* be large enough to learn such an approximately-optimal contract. To circumvent Theorem 1, in the rest of the paper we focus on designing algorithms for the problem introduced in the following Definition 1. Such a problem relaxes the one introduced above by asking to find a contract whose seller's expected utility is "approximately" close to that of a utility-maximizing contract among those with *bounded* payments.

**Definition 1** (Learning an optimal bounded contract)**.** *The problem of* learning an optimal bounded contract *reads as follows: Given a bound $B \geq 1$ on payments, an additive approximation error $\rho > 0$,*

---

[3]Such a learning goal can be seen as instantiating the PAC-learning framework into principal-agent settings.

*and a probability $\delta \in (0, 1)$), find a contract $p \in [0, B]^m$ such that the following holds:*

$$\mathbb{P}\left\{u(p) \geq \max_{p' \in [0,B]^m} u(p') - \rho\right\} \geq 1 - \delta.$$

Let us remark that the problem introduced in Definition 1 does *not* substantially hinder the generality of the problem of learning an optimal contract. Indeed, since a contract achieving principal's expected utility OPT can be found by means of linear programming (Dutting et al., 2021), it is always the case that an optimal contract uses payments which can be bounded above in terms of the number of bits required to represent the probability distributions over outcomes. Moreover, all the previous works that study the sample complexity of learning optimal contracts focus on settings with bounded payments; see, *e.g.*, (Zhu et al., 2023). The latter only considers contracts in $[0, 1]^m$, while we address the more general case in which the payments are bounded above by a given $B \geq 1$.

## 4 THE DISCOVER-AND-COVER ALGORITHM

In this section, we provide an algorithm for the problem introduced in Definition 1, which we call `Discover-and-Cover` algorithm (Algorithm 1). Our main result (Theorem 2) is an upper bound on the number of rounds required by the algorithm, which we show to grow polynomially in the size of the problem instance when the number of agent's actions $n$ is constant.

The core idea behind Algorithm 1 is to learn an optimal bounded contract $p \in [0, B]^m$ by "approximately" identifying the best-response regions $\mathcal{P}_a$. Notice that, since at the end of each round the principal only observes the outcome realized by the agent's action, rather than the action itself, identifying such best-response regions exactly is *not* possible. Moreover, the principal does *not* even know the set of actions available to the agent, which makes the task even more challenging.

Algorithm 1 builds a set $\mathcal{D}$ of *meta-actions*, where each meta-action $d \in \mathcal{D}$ identifies a set $A(d) \subseteq \mathcal{A}$ of one or more (unknown) agent's actions. A meta-action groups together "similar" actions, in the sense that they induce similar distributions over outcomes. The algorithm incrementally discovers new elements of $\mathcal{D}$ by trying to cover a suitably-defined set $\mathcal{P}$ of contracts (see Line 2) by means of *approximate best-response regions* $\mathcal{L}_d$, one for each $d \in \mathcal{D}$. In particular, $\mathcal{L}_d$ is a suitably-defined polytope that "approximately describes" the union of all the best-response regions $\mathcal{P}_a$ of actions $a \in A(d)$ associated with

---

**Algorithm 1** `Discover-and-Cover`

**Require:** $\rho \in (0, 1), \delta \in (0, 1), B \geq 1$
1: Set $\epsilon, \alpha, q$, and $\eta$ as in Appendix C.1
2: $\mathcal{P} \leftarrow \{p \in \mathbb{R}_+^m \mid \|p\|_1 \leq mB\}$
3: $\mathcal{D} \leftarrow \varnothing, \mathcal{F} \leftarrow \varnothing$
4: **do**
5: $\quad \{\mathcal{L}_d\}_{d \in \mathcal{D}} \leftarrow$ `Try-Cover()`
6: **while** $\{\mathcal{L}_d\}_{d \in \mathcal{D}} = \varnothing$
7: **return** `Find-Contract`$(\{\mathcal{L}_d\}_{d \in \mathcal{D}})$

---

$d$. Each time $\mathcal{D}$ is updated, the algorithm calls the `Try-Cover` procedure (Algorithm 3), whose aim is to cover $\mathcal{P}$ with approximate best-response regions. `Try-Cover` works by iteratively finding hyperplanes that separate such regions, by "testing" suitable contracts through the `Action-Oracle` procedure (Algorithm 2). Given a contract $p \in \mathcal{P}$, `Action-Oracle` checks whether it is possible to safely map the agent's best response $a^\star(p)$ to an already-discovered meta-action $d \in \mathcal{D}$ or *not*. In the latter case, `Action-Oracle` refines the set $\mathcal{D}$ by either adding a new meta-action or merging a group of meta-actions if their associated actions can be considered as "similar". Whenever the set $\mathcal{D}$ changes, `Try-Cover` is terminated (returning $\varnothing$). The algorithm continues trying to cover $\mathcal{P}$ until a satisfactory cover is found. Finally, such a cover is used to compute a contract $p \in [0, B]^m$ to be returned, by means of the procedure `Find-Contract` (Algorithm 4).

In the rest of this section, we describe the details of all the procedures used by Algorithm 1. In particular, Section 4.1 is concerned with `Action-Oracle`, Section 4.2 with `Try-Cover`, and Section 4.3 with `Find-Contract`. Finally, Section 4.4 concludes the analysis of Algorithm 1 by putting everything together so as to prove the guarantees of the algorithm.

### 4.1 ACTION-ORACLE

Given a contract $p \in \mathcal{P}$ as input, the `Action-Oracle` procedure (Algorithm 2) determines whether it is possible to safely map the agent's best response $a^\star(p)$ to some meta-action $d \in \mathcal{D}$. If this is the case, then the procedure returns such a meta-action. Otherwise, the procedure has to properly refine the set $\mathcal{D}$ of meta-actions, as we describe in the rest of this subsection.

In the first phase (Lines 1–6), Algorithm 2 prescribes the principal to commit to the contract $p \in \mathcal{P}$ given as input for $q \in \mathbb{N}$ consecutive rounds (see Appendix C.1 for the definition of $q$). This is done to build an empirical distribution $\widetilde{F} \in \Delta_\Omega$ that estimates the (true) distribution over outcomes $F_{a^\star(p)}$ induced by the agent's best response $a^\star(p)$. In the second phase (Lines 7–21), such an empirical distribution is compared with those previously computed, in order to decide whether $a^\star(p)$ can be safely mapped to some $d \in \mathcal{D}$ or *not*. To perform such a comparison, all the empirical distributions computed by the algorithm are stored in a dictionary $\mathcal{F}$ (initialized in Line 3 of Algorithm 1), where $\mathcal{F}[d]$ contains all the $\widetilde{F} \in \Delta_\Omega$ associated with $d \in \mathcal{D}$.

The first phase of Algorithm 2 ensures that $\|\widetilde{F} - F_{a^\star(p)}\|_\infty \leq \epsilon$ holds with sufficiently high probability. For ease of presentation, we introduce a *clean event* that allows us to factor out from our analysis the "with high probability" statements. In Section 4.4, we will show that such an event holds with sufficiently high probability given how the value of $q$ is chosen.

**Definition 2** (Clean event). *Given any $\epsilon > 0$, we let $\mathcal{E}_\epsilon$ be the event in which Algorithm 2 always computes some $\widetilde{F} \in \Delta_\Omega$ such that $\|\widetilde{F} - F_{a^\star(p)}\|_\infty \leq \epsilon$, where $p \in \mathcal{P}$ is the input to the algorithm.*

In the second phase, Algorithm 2 searches for all the $d \in \mathcal{D}$ such that $\mathcal{F}[d]$ contains *at least one* empirical distribution which is "sufficiently close" to the $\widetilde{F}$ that has just been computed by the algorithm. Formally, the algorithm looks for all the meta-actions $d \in \mathcal{D}$ such that $\|F - \widetilde{F}\|_\infty \leq 2\epsilon$ for at least one $F \in \mathcal{F}[d]$. Then, three cases are possible: (i) If the algorithm finds a *unique* $d \in \mathcal{D}$ with such a property (case $|\mathcal{D}^\diamond| = 1$), then $d$ is returned since $a^\star(p)$ can be safely mapped to $d$. (ii) If the algorithm does *not* find any $d \in \mathcal{D}$ with such a property (case $\mathcal{D}^\diamond = \varnothing$), then a new meta-action $d^\diamond$ is added to $\mathcal{D}$. (iii) If the algorithm finds *more than one* $d \in \mathcal{D}$ with such a property ($|\mathcal{D}^\diamond| > 1$), then all the meta-actions in $\mathcal{D}^\diamond$ are merged into a single (new) meta-action $d^\diamond$. In Algorithm 2, the last two cases above are jointly managed by Lines 17–21, and in both cases the algorithm returns the special value $\bot$. This alerts the calling procedure that the set $\mathcal{D}$ has changed, and, thus, the `Try-Cover` procedure needs to be re-started. Notice that Algorithm 2 also takes care of properly updating the dictionary $\mathcal{F}$, which is done in Lines 14 and 20. Moreover, whenever Algorithm 2 adds a new meta-action $d^\diamond$ into $\mathcal{D}$, this is also associated with

---

**Algorithm 2** `Action-Oracle`

---

**Require:** $p \in \mathcal{P}$
1: $I_\omega \leftarrow 0$ for all $\omega \in \Omega$ ▷ Phase 1: Estimate
2: **for** $\tau = 1, \dots, q$ **do**
3:     Commit to $p$ and observe $\omega \in \Omega$
4:     $I_\omega \leftarrow I_\omega + 1$
5: **end for**
6: Build $\widetilde{F} \in \Delta_\Omega : \widetilde{F}_\omega = I_\omega / q \; \forall \omega \in \Omega$
7: $\mathcal{D}^\diamond \leftarrow \varnothing$          ▷ Phase 2: Compare
8: **for** $d \in \mathcal{D}$ **do**
9:     **if** $\exists F \in \mathcal{F}[d] : \|F - \widetilde{F}\|_\infty \leq 2\epsilon$ **then**
10:         $\mathcal{D}^\diamond \leftarrow \mathcal{D}^\diamond \cup \{d\}$
11:     **end if**
12: **end for**
13: **if** $|\mathcal{D}^\diamond| = 1$ **then**
14:     $\mathcal{F}[d] \leftarrow \mathcal{F}[d] \cup \{\widetilde{F}\}$ ▷ unique $d \in \mathcal{D}^\diamond$
15:     **return** $d$
16: **end if**
17: $\mathcal{D} \leftarrow (\mathcal{D} \setminus \mathcal{D}^\diamond) \cup \{d^\diamond\}$     ▷ $d^\diamond$ is new
18: $\mathcal{F}[d^\diamond] \leftarrow \bigcup_{d \in \mathcal{D}^\diamond} \mathcal{F}[d] \cup \{\widetilde{F}\}$
19: Let $\widetilde{F}_{d^\diamond}$ be equal to $\widetilde{F}$
20: Clear $\mathcal{F}[d]$ for all $d \in \mathcal{D}^\diamond$
21: **return** $\bot$

---

a particular empirical distribution $\widetilde{F}_{d^\diamond}$, which is set to be equal to $\widetilde{F}$ (Line 19). We remark that the symbols $\widetilde{F}_d$ for $d \in \mathcal{D}$ have only been introduced for ease of exposition, and they do *not* reference actual variables declared in the algorithm. Operationally, one can replace each appearance of $\widetilde{F}_d$ in the algorithms with any fixed empirical distribution contained in the dictionary entry $\mathcal{F}[d]$.

The first crucial property that Algorithm 2 guarantees is that *only "similar" agent's best-response actions are mapped to the same meta-action*. In order to formally state such a property, we first need to introduce the definition of set of agent's actions *associated with* a meta-action in $\mathcal{D}$.

**Definition 3** (Associated actions). *Given a set $\mathcal{D}$ of meta-actions and a dictionary $\mathcal{F}$ of empirical distributions computed by means of Algorithm 2, we let $A : \mathcal{D} \to 2^{\mathcal{A}}$ be a function such that $A(d)$ represents the set of actions associated with the meta-action $d \in \mathcal{D}$, defined as follows:*

$$A(d) = \left\{ a \in \mathcal{A} \mid \|\widetilde{F}_d - F_a\|_\infty \leq 5\epsilon n \wedge \exists p \in \mathcal{P} : a = a^\star(p) \right\}.$$

The set $A(d)$ encompasses all the agent's actions $a \in \mathcal{A}$ whose distributions over outcomes $F_a$ are "similar" to the empirical distribution $\widetilde{F}_d$ associated with the meta-action $d \in \mathcal{D}$, where the entity of the similarity is defined in a suitable way depending on $\epsilon$ and the number of agent's actions $n$. Moreover, notice that the set $A(d)$ only contains agent's actions that can be induced as best response for at least one contract $p \in \mathcal{P}$. This is needed in order to simplify the analysis of the algorithm. Let

us also remark that an agent's action may be associated with *more than one* meta-action. Equipped with Definition 3, the property introduced above can be formally stated as follows:

**Lemma 1.** *Given a set $\mathcal{D}$ of meta-actions and a dictionary $\mathcal{F}$ of empirical distributions computed by means of Algorithm 2, under the event $\mathcal{E}_\epsilon$, if Algorithm 2 returns a meta-action $d \in \mathcal{D}$ for a contract $p \in \mathcal{P}$ given as input, then it holds that $a^\star(p) \in A(d)$.*

Lemma 1 follows from the observation that, under the event $\mathcal{E}_\epsilon$, the empirical distributions computed by Algorithm 2 are "close" to the true ones, and, thus, the distributions over outcomes of all the actions associated with $d$ are sufficiently "close" to each other. A non-trivial part of the proof of Lemma 1 is to show that Algorithm 2 does *not* put in the same entry $\mathcal{F}[d]$ empirical distributions that form a "chain" growing arbitrarily in terms of $\|\cdot\|_\infty$ norm, but instead the length of such "chains" is always bounded by $5\epsilon n$. The second crucial property is made formal by the following lemma:

**Lemma 2.** *Under the event $\mathcal{E}_\epsilon$, Algorithm 2 returns $\perp$ at most $2n$ times.*

Intuitively, Lemma 2 follows from the fact that Algorithm 2 increases the cardinality of the set $\mathcal{D}$ by one only when $\|F - \widetilde{F}\|_\infty > 2\epsilon$ for all $F \in \mathcal{F}[d]$ and $d \in \mathcal{D}$. In such a case, under the event $\mathcal{E}_\epsilon$ the agent's best response is an action that has never been played before. Thus, the cardinality of $\mathcal{D}$ can be increased at most $n$ times. Moreover, in the worst case the cardinality of $\mathcal{D}$ is reduced by one for $n$ times, resulting in $2n$ being the maximum number of times Algorithm 2 returns $\perp$. In the following, we formally introduce the definition of *cost of a meta-action*.

**Definition 4** (Cost of a meta-action). *Given a set $\mathcal{D}$ of meta-actions and a dictionary $\mathcal{F}$ of empirical distributions computed by means of Algorithm 2, we let $c : \mathcal{D} \to [0, 1]$ be a function such that $c(d)$ represents the* cost of meta-action $d \in \mathcal{D}$, *defined as $c(d) = \min_{a \in A(d)} c_a$.*

Then, by Holder's inequality, we can prove that the two following lemmas hold:

**Lemma 3.** *Given a set $\mathcal{D}$ of meta-actions and a dictionary $\mathcal{F}$ of empirical distributions computed by means of Algorithm 2, under the event $\mathcal{E}_\epsilon$, for every meta-action $d \in \mathcal{D}$ and associated action $a \in A(d)$ it holds that $|c(d) - c_a| \leq 4B\epsilon mn$.*

**Lemma 4.** *Given a set $\mathcal{D}$ of meta-actions and a dictionary $\mathcal{F}$ of empirical distributions computed by means of Algorithm 2, under the event $\mathcal{E}_\epsilon$, for every meta-action $d \in \mathcal{D}$, associated action $a \in A(d)$, and contract $p \in \mathcal{P}$ it holds $|\sum_{\omega \in \Omega} \widetilde{F}_{d,\omega} \, p_\omega - c(d) - \sum_{\omega \in \Omega} F_{a,\omega} \, p_\omega + c_a| \leq 9B\epsilon mn$.*

## 4.2 TRY-COVER

In this section, we present key component of Algorithm 1, which is the `Try-Cover` procedure (Algorithm 3). It builds a cover $\{\mathcal{L}_d\}_{d \in \mathcal{D}}$ of $\mathcal{P}$ made of approximate best-response regions for the meta-actions in the current set $\mathcal{D}$. The approximate best-response region $\mathcal{L}_d$ for a meta-action $d \in \mathcal{D}$ must be such that: (i) all the best-response regions $\mathcal{P}_a$ of actions $a \in A(d)$ associated with $d$ are contained in $\mathcal{L}_d$; and (ii) for every contract in $\mathcal{L}_d$ there must be an action $a \in A(d)$ that is induced as an "approximate best response" by that contract. Notice that working with approximate best-response regions is unavoidable since the algorithm has only access to estimates of the (true) distributions over outcomes induced by agent's actions.

The core idea of the algorithm is to progressively build two polytopes $\mathcal{U}_d$ and $\mathcal{L}_d$—called, respectively, *upper bound* and *lower bound*—for each meta-action $d \in \mathcal{D}$. During the execution, the upper bound $\mathcal{U}_d$ is continuously shrunk in such a way that a suitable approximate best-response region for $d$ is guaranteed to be contained in $\mathcal{U}_d$, while the lower bound $\mathcal{L}_d$ is progressively expanded so that it is contained in such a region. The algorithm may terminate in two different ways. The first one is when $\mathcal{L}_d = \mathcal{U}_d$ for every $d \in \mathcal{D}$. In such a case, the lower bounds $\{\mathcal{L}_d\}_{d \in \mathcal{D}}$ constitutes a cover of $\mathcal{P}$ made of suitable approximate best-response regions for the meta-actions in $\mathcal{D}$, and, thus, it is returned by the algorithm. The second way of terminating occurs any time a call to the `Action-Oracle` procedure is *not* able to safely map the agent's best response under the contract given as input to a meta-action (*i.e.*, `Action-Oracle` returns $\perp$). In such a case, the algorithm gives back control to Algorithm 1 by returning $\varnothing$, and the latter in turn re-starts the `Try-Cover` procedure from scratch since the last call to `Action-Oracle` has updated $\mathcal{D}$. This happens in Lines 5, 14, and 22.

The ultimate goal of Algorithm 3 is to reach termination with $\mathcal{L}_d = \mathcal{U}_d$ for every $d \in \mathcal{D}$. Intuitively, the algorithm tries to "close the gap" between the lower bound $\mathcal{L}_d$ and the upper bound $\mathcal{U}_d$ for each $d \in$

$\mathcal{D}$ by discovering suitable *approximate halfspaces* whose intersections define the desired approximate best-response regions. Such halfspaces are found in Line 21 by means of the `Find-HS` procedure (Algorithm 6), whose description and analysis is deferred to Appendix B. Intuitively, such a procedure searches for an hyperplane that defines a suitable approximate halfspace by performing a binary search (with parameter $\eta$) on the line connecting two given contracts, by calling `Action-Oracle` on the contract representing the middle point of the line at each iteration.

The halfspaces that define the approximate best-response regions computed by Algorithm 3 intuitively identify areas of $\mathcal{P}$ in which one meta-action is "supposedly better" than another one. In particular, Algorithm 3 uses the variable $\widetilde{H}_{ij}$ to store the approximate halfspace in which $d_i \in \mathcal{D}$ is "supposedly better" than $d_j \in \mathcal{D}$, in the sense that, for every contract in such a halfspace, each action associated with $d_i$ provides (approximately) higher utility to the agent than all the actions associated with $d_j$. Then, the approximate best-response region for $d_i \in \mathcal{D}$ is built by intersecting a suitably-defined group of $\widetilde{H}_{ij}$, for some $d_j \in \mathcal{D}$ with $j \neq i$. In order to ease the construction of the approximate halfspaces in the `Find-HS` procedure, Algorithm 3 also keeps some variables $\Delta\widetilde{c}_{ij}$, which represent estimates of the difference between the costs $c(d_i)$ and $c(d_j)$ of $d_i$ and $d_j$, respectively. These are needed to easily compute the intercept values of the hyperplanes defining $\widetilde{H}_{ij}$.

Next, we describe the procedure used by Algorithm 3 to reach the desired termination. In the following, for clarity of exposition, we omit all the cases in which the algorithm ends prematurely after a call to `Action-Oracle`. Algorithm 3 works by tracking all the $d \in \mathcal{D}$ that still need to be processed, *i.e.*, those such that $\mathcal{L}_d \neq \mathcal{U}_d$, into a set $\mathcal{C}$. At the beginning of the algorithm (Lines 1–6), all the variables are properly initialized. In particular, all the upper bounds are initialized to the set $\mathcal{P}$, while all the lower bounds are set to $\varnothing$. The set $\mathcal{C}$ is initialized to contain a $d \in \mathcal{D}$ obtained by calling `Action-Oracle` for any contract $p \in \mathcal{P}$, with the lower bound $\mathcal{L}_d$ being updated to the singleton $\{p\}$.

**Algorithm 3** `Try-Cover`

1: $\mathcal{L}_d \leftarrow \varnothing, \mathcal{U}_d \leftarrow \mathcal{P}$ for all $d \in \mathcal{D}$
2: $\mathcal{D}_d \leftarrow \{d\}$ for all $d \in \mathcal{D}$
3: $\widetilde{H}_{ij} \leftarrow \varnothing, \Delta\widetilde{c}_{ij} \leftarrow 0$ for all $d_i, d_j \in \mathcal{D}$
4: $d \leftarrow$ `Action-Oracle`$(p)$ ▷ For any $p$
5: **if** $d = \bot$ **then return** $\varnothing$
6: $\mathcal{L}_d \leftarrow \{p\}, \mathcal{C} \leftarrow \{d\}$
7: **while** $\mathcal{C} \neq \varnothing$ **do**
8:     Take any $d_i \in \mathcal{C}$
9:     **while** $\mathcal{L}_{d_i} \neq \mathcal{U}_{d_i}$ **do**
10:         $V_{d_i} \leftarrow V(\mathcal{U}_{d_i})$ ▷ Vertexes of $\mathcal{U}_{d_i}$
11:         **do**
12:             Take any $p \in V_{d_i}$
13:             $d_j \leftarrow$ `Action-Oracle`$(p)$
14:             **if** $d_j = \bot$ **then return** $\varnothing$
15:             **if** $\mathcal{L}_{d_j} = \varnothing$ **then**
16:                 $\mathcal{L}_{d_j} \leftarrow \{p\}, \mathcal{C} \leftarrow \mathcal{C} \cup \{d_j\}$
17:             **end if**
18:             **if** $d_j \in \mathcal{D}_{d_i}$ **then**
19:                 $\mathcal{L}_{d_i} \leftarrow \text{co}(\mathcal{L}_{d_i}, p)$
20:             **else**
21:                 $(\widetilde{H}, d_k) \leftarrow$ `Find-HS`$(d_i, p)$
22:                 **if** $d_k = \bot$ **then return** $\varnothing$
23:                 $\widetilde{H}_{ik} \leftarrow \widetilde{H}$
24:                 $\mathcal{U}_{d_i} \leftarrow \mathcal{U}_{d_i} \cap \widetilde{H}_{ik}$
25:                 $\mathcal{D}_{d_i} \leftarrow \mathcal{D}_{d_i} \cup \{d_k\}$
26:             **end if**
27:             $V_{d_i} \leftarrow V_{d_i} \setminus \{p\}$
28:         **while** $V_{d_i} \neq \varnothing \wedge d_j \in \mathcal{D}_{d_i}$
29:     **end while**
30:     $\mathcal{C} \leftarrow \mathcal{C} \setminus \{d_i\}$
31: **end while**
32: **return** $\{\mathcal{L}_d\}_{d \in \mathcal{D}}$

Moreover, Algorithm 3 also maintains some subsets $\mathcal{D}_d \subseteq \mathcal{D}$ of meta-actions, one for each $d \in \mathcal{D}$. For any $d_i \in \mathcal{D}$, the set $\mathcal{D}_{d_i}$ is updated so as to contain all the $d_j \in \mathcal{D}$ such that the halfspace $\widetilde{H}_{ij}$ has already been computed. Each set $\mathcal{D}_d$ is initialized to be equal to $\{d\}$, as this is useful to simplify the pseudocode of the algorithm. The main loop of the algorithm (Line 7) iterates over the meta-actions in $\mathcal{C}$ until such a set becomes empty. For every $d_i \in \mathcal{C}$, the algorithm tries to "close the gap" between the lower bound $\mathcal{L}_{d_i}$ and the upper bound $\mathcal{U}_{d_i}$ by using the loop in Line 9 In particular, the algorithm does so by iterating over all the vertices $V(\mathcal{U}_{d_i})$ of the polytope defining the upper bound $\mathcal{U}_{d_i}$ (see Line 10). For every $p \in V(\mathcal{U}_{d_i})$, the algorithm first calls `Action-Oracle` with $p$ as input. Then:

- If the returned $d_j$ belongs to $\mathcal{D}_{d_i}$, then the algorithm takes the convex hull between the current lower bound $\mathcal{L}_{d_i}$ and $p$ as a new lower bound for $d_i$. This may happen when either $d_j = d_i$ (recall that $d_i \in \mathcal{D}_{d_i}$ by construction) or $d_j \neq d_i$. Intuitively, in the former case the lower bound can be "safely expanded" to match the upper bound at the currently-considered vertex $p$, while in the latter case such "matching" the upper bound may introduce additional errors in the definition of $\mathcal{L}_{d_i}$. Nevertheless, such an operation is done in both cases, since this *not* hinders the guarantees of the algorithm, as we formally show in Lemma 6. Notice that handling the case $d_j \neq d_i$ as in Algorithm 3 is crucial to avoid that multiple versions of the approximate halfspace $\widetilde{H}_{ij}$ are created during the execution of the algorithm.

- If the returned $d_j$ does *not* belong to $\mathcal{D}_{d_i}$, the algorithm calls the `Find-HS` procedure to find a new approximate halfspace. Whenever the procedure is successful, it returns an approximate

halfspace $\widetilde{H}$ that identifies an area of $\mathcal{P}$ in which $d_i$ is "supposedly better" than another meta-action $d_k \in \mathcal{D} \setminus \mathcal{D}_{d_i}$, which is returned by the Find-HS procedure as well. Then, the returned halfspace $\widetilde{H}$ is copied into the variable $\widetilde{H}_{ik}$ (Line 23), and the upper bound $\mathcal{U}_{d_i}$ is intersected with the latter (Line 24). Moreover, $d_k$ is added to $\mathcal{D}_{d_i}$ (Line 25), in order to record that the halfspace $\widetilde{H}_{ik}$ has been found. After that, the loop over vertexes is re-started, as the upper bound has been updated.

Whenever the loop in Line 9 terminates, $\mathcal{L}_{d_i} = \mathcal{U}_{d_i}$ for the current meta-action $d_i$, and, thus, $d_i$ is removed from $\mathcal{C}$. Moreover, if a call to Action-Oracle returns a meta-action $d_j \in \mathcal{D}$ such that $d_j \notin \mathcal{C}$ and $\mathcal{L}_{d_j} = \varnothing$, then $d_j$ is added to $\mathcal{C}$ and $\mathcal{L}_{d_j}$ is set to $\{p\}$, where $p \in \mathcal{P}$ is the contract given to Action-Oracle (see Lines 15–16). This ensures that all the meta-actions are eventually considered. Next, we prove two crucial properties which are satisfied by Algorithm 3 whenever it returns $\{\mathcal{L}_d\}_{d \in \mathcal{D}}$. The first one is formally stated in the following lemma:

**Lemma 5.** *Under the event $\mathcal{E}_\epsilon$, when Algorithm 3 returns $\{\mathcal{L}_d\}_{d \in \mathcal{D}}$, it holds that $\bigcup_{d \in \mathcal{D}} \mathcal{L}_d = \mathcal{P}$.*

Intuitively, Lemma 5 states that Algorithm 3 terminates with a correct cover of $\mathcal{P}$, and it follows from the fact that, at the end of the algorithm, $\bigcup_{d \in \mathcal{D}} \mathcal{U}_d = \mathcal{P}$ and $\mathcal{L}_d = \mathcal{U}_d$ for every $d \in \mathcal{D}$. The second crucial lemma states the following:

**Lemma 6.** *Under the event $\mathcal{E}_\epsilon$, when Algorithm 3 returns $\{\mathcal{L}_d\}_{d \in \mathcal{D}}$, for every meta-action $d \in \mathcal{D}$, contract $p \in \mathcal{L}_d$ and action $a' \in A(d)$, there exists a $\gamma$ that polynomially depends on $m$, $n$, $\epsilon$, and $B$ such that:*

$$\sum_{\omega \in \Omega} F_{a',\omega} \, p_\omega - c_{a'} \geq \sum_{\omega \in \Omega} F_{a,\omega} \, p_\omega - c_a - \gamma \quad \forall a \in \mathcal{A}.$$

Lemma 6 states that $\{\mathcal{L}_d\}_{d \in \mathcal{D}}$ defines a cover of $\mathcal{P}$ made of suitable approximate best-response regions for the meta-actions in $\mathcal{D}$. Indeed, for every meta-action $d \in \mathcal{D}$ and contract $p \in \mathcal{L}_d$, playing any $a \in A(d)$ associated with $d$ is an "approximate best response" for the agent, in the sense that the agent's utility only decreases by a small amount with respect to playing the (true) best response $a^\star(p)$. Finally, the following lemma bounds the number of rounds required by Algorithm 3.

**Lemma 7.** *Under event $\mathcal{E}_\epsilon$, Algorithm 3 requires at most $\mathcal{O}\left(n^2 q \left(\log\left(\frac{Bm}{\eta}\right) + \binom{m+n+1}{m}\right)\right)$ rounds.*

Lemma 7 follows from the observation that the main cost, in terms of number of rounds, incurred by Algorithm 3 is to check all the vertexes of the upper bounds $\mathcal{U}_d$. The number of such vertexes can be bound by $\binom{n+m+1}{m}$, thanks to the fact that the set $\mathcal{P}$ being covered by Algorithm 3 has $m+1$ vertexes. Notice that, using $\mathcal{P}$ instead of $[0, B]^m$ is necessary, since the latter has a number vertexes which is exponential in $m$. Nevertheless, Algorithm 1 returns a contract $p \in [0, B]^m$ by means of the Find-Contract procedure, which we are going to describe in the following subsection.

## 4.3 FIND-CONTRACT

The Find-Contract procedure (Algorithm 4) finds a contract $p \in [0, B]^m$ that approximately maximizes the principal's expected utility over $[0, B]^m$ by using the cover $\{\mathcal{L}_d\}_{d \in \mathcal{D}}$ of $\mathcal{P}$ made by approximate best-response regions given as input (obtained by running Try-Cover).

First, for every $d \in \mathcal{D}$, Algorithm 4 computes $p_d^\star$ which maximizes an empirical estimate of the principal's expected utility over the polytope $\mathcal{L}_d \cap [0, B]^m$. This is done in Line 2 by solving a *linear program* with constraints defined by the hyperplanes identifying $\mathcal{L}_d \cap [0, B]^m$ and objective function defined by the principal's expected utility when outcomes are generated by $\widetilde{F}_d$. Then, Algorithm 4 takes the best contract (according to the empirical distributions $\widetilde{F}_d$) among all the $p_d^\star$, which is the contract $p^\star$ defined in Line 5, and it returns a suitable convex combination of such a contract and a vector whose components are defined

---

**Algorithm 4** Find-Contract

**Require:** $\{\mathcal{L}_d\}_{d \in \mathcal{D}}$
1: **while** $d \in \mathcal{D}$ **do**
2: $\qquad p_d^\star \leftarrow \arg\max_{p \in [0,B]^m \cap \mathcal{L}_d} \sum_{\omega \in \Omega} \widetilde{F}_{d,\omega} \left(r_\omega - p_\omega\right)$
3: **end while**
4: $d^\star \leftarrow \arg\max_{d \in \mathcal{D}} \sum_{\omega \in \Omega} \widetilde{F}_{d,\omega} (r_\omega - p_{d,\omega}^\star)$
5: $p^\star \leftarrow p_{d^\star}^\star$
6: $p_\omega \leftarrow (1 - \sqrt{\epsilon}) p_\omega^\star + \sqrt{\epsilon} r_\omega$ for all $\omega \in \Omega$
7: **return** $p$

---

by principal's rewards (see Line 6). The following lemma formally proves the guarantees in terms of principal's expected utility provided by Algorithm 4:

**Lemma 8.** *Under the event $\mathcal{E}_\epsilon$, if $\{\mathcal{L}_d\}_{d \in \mathcal{D}}$ is a cover of $\mathcal{P}$ computed by Try-Cover, Algorithm 4 returns a contract $p \in [0, B]^m$ such that $u(p) \geq \max_{p' \in [0,B]^m} u(p') - \rho$.*

The main challenge in proving Lemma 8 is that, for the contract $p^\star$ computed in Line 5, the agent's best response may *not* be associated with any meta-action in $\mathcal{D}$, namely $a^\star(p^\star) \notin A(d)$ for every $d \in \mathcal{D}$. Nevertheless, by means of Lemma 6, we can show that $a^\star(p^\star)$ is an approximate best response to $p^\star$. Moreover, the algorithm returns the contract defined in Line 6, *i.e.*, a convex combination of $p^\star$ and the principal's reward vector. Intuitively, paying the agent based on the principal's rewards aligns the agent's interests with those of the principal. This converts the approximate incentive compatibility of $a^\star(p^\star)$ for the contract $p^\star$ into a loss in terms of principal's expected utility.

## 4.4 PUTTING IT ALL TOGETHER

We conclude the section by putting all the results related to the procedures involved in Algorithm 1 together, in order to derive the guarantees of the algorithm.

First, by Lemma 2 the number of calls to the `Try-Cover` procedure is at most $2n$. Moreover, by Lemma 7 and by definition of $q$ and $\eta$, the number of rounds required by each call to `Try-Cover` is at most $\widetilde{\mathcal{O}}(m^n \cdot \mathcal{I} \cdot {}^1\!/_{\rho^4} \log({}^1\!/_\delta))$ under the event $\mathcal{E}_\epsilon$, where $\mathcal{I}$ is a term that depends polynomially in $m$, $n$, and $B$. Finally, by Lemma 8 the contract returned by Algorithm 1—the result of a call to the `Find-Contract` procedure—has expected utility at most $\rho$ less than the best contact in $[0, B]^m$, while the probability of the clean event $\mathcal{E}_\epsilon$ can be bounded below by means of a concentration argument. All the observations above allow us to state the following main result.

**Theorem 2.** *Given $\rho \in (0, 1)$, $\delta \in (0, 1)$, and $B \geq 1$ as inputs, with probability at least $1 - \delta$ the* `Discover-and-Cover` *algorithm (Algorithm 1) is guaranteed to return a contract $p \in [0, B]^m$ such that $u(p) \geq \max_{p' \in [0,B]^m} u(p') - \rho$ in at most $\widetilde{\mathcal{O}}(m^n \cdot \mathcal{I} \cdot {}^1\!/_{\rho^4} \log({}^1\!/_\delta))$ rounds, where $\mathcal{I}$ is a term that depends polynomially in $m$, $n$, and $B$.*

Notice that the number of rounds required by Algorithm 1 is polynomial in the instance size (including the number of outcomes $m$) when the number of agent's actions $n$ is constant.

## 5 CONNECTION WITH ONLINE LEARNING IN PRINCIPAL-AGENT PROBLEMS

In this section, we show that our `Discover-and-Cover` algorithm can be exploited to derive a no-regret algorithm for the related online learning setting in which the principal aims at maximizing their cumulative utility. In such a setting, the principal and the agent interact repeatedly over a given number of rounds $T$, as described in Section 3.

At each $t = 1, \ldots, T$, the principal commits to a contract $p^t \in \mathbb{R}_+^m$, the agent plays a best response $a^\star(p^t)$, and the principal observes the realized outcome $\omega^t \sim F_{a^\star(p^t)}$ with reward $r_{\omega^t}$. Then, the performance in terms of cumulative expected utility by employing the contracts $\{p^t\}_{t=1}^T$ is measured by the *cumulative (Stackelberg) regret*

$$R^T := T \cdot \max_{p \in [0,B]^m} u(p) - \mathbb{E}\left[\sum_{t=1}^T u(p^t)\right],$$

where the expectation is over the randomness of the environment. As shown in Section 4, Algorithm 1

---

**Algorithm 5** No-regret algorithm
<hr/>

**Require:** $\delta \in (0, 1)$, $B \geq 1$
1: Set $\rho$ as in proof of Theorem 3
2: **for** $t = 1, \ldots, T$ **do**
3:     **if** Algorithm 1 *not* terminated yet **then**
4:         $p^t \leftarrow p \in \mathcal{P}$ prescribed by Alg. 1
5:     **else**
6:         $p^t \leftarrow p \in [0, B]^m$ returned by Alg. 1
7:     **end if**
8: **end for**

---

learns an approximately-optimal bounded contract with high probability by using the number of rounds prescribed by Theorem 2. Thus, by exploiting Algorithm 1, it is possible to design an *explore-then-commit* algorithm ensuring sublinear regret $R^T$ with high probability; see Algorithm 5.

**Theorem 3.** *Given $\alpha \in (0, 1)$, Algorithm 5 achieves $R^T \leq \widetilde{\mathcal{O}}\left(m^n \cdot \mathcal{I} \cdot \log({}^1\!/_\delta) \cdot T^{4/5}\right)$ with probability at least $1 - \delta$, where $\mathcal{I}$ is a term that depends polynomially on $m$, $n$, and $B$.*

Notice that, even for a small number of outcomes $m$ (*i.e.*, any $m \geq 3$), our algorithm achieves better regret guarantees than those obtained by Zhu et al. (2023) in terms of the dependency on the number of rounds $T$. Specifically, Zhu et al. (2023) provide a $\widetilde{\mathcal{O}}(T^{1-1/(2m+1)})$ regret bound, which exhibits a very unpleasant dependency on the number of outcomes $m$ at the exponent of $T$. Conversely, our algorithm always achieves a $\widetilde{\mathcal{O}}(T^{4/5})$ dependency on $T$, when the number of agent's actions $n$ is small. This solves a problem left open in the very recent paper by Zhu et al. (2023).

ACKNOWLEDGMENTS

This paper is supported by the Italian MIUR PRIN 2022 Project "Targeted Learning Dynamics: Computing Efficient and Fair Equilibria through No-Regret Algorithms", by the FAIR (Future Artificial Intelligence Research) project, funded by the NextGenerationEU program within the PNRR-PE-AI scheme (M4C2, Investment 1.3, Line on Artificial Intelligence), and by the EU Horizon project ELIAS (European Lighthouse of AI for Sustainability, No. 101120237).

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

APPENDIX

The appendixes are organized as follows:

- Appendix A provides a detailed discussion of the previous works most related to ours.
- Appendix B provides all the details about the `Find-HS` procedure which is employed by the `Discover-and-Cover` algorithm, including all the technical lemmas (and their proofs) related to such a procedure.
- Appendix C provides all the details about the `Discover-and-Cover` algorithm that are omitted from the main body of the paper, including the definitions of the parameters required by the algorithm and the proofs of all the results related to it.
- Appendix D provides all the other proofs omitted from the main body of the paper.

## A  RELATED WORKS

In this section, we survey all the previous works that are most related to ours. Among the works addressing principal-agent problems, we only discuss those focusing on learning aspects. Notice that there are several works studying computational properties of principal-agent problems which are *not* concerned with learning aspects, such as, *e.g.*, (Dütting et al., 2019; Alon et al., 2021; Guruganesh et al., 2021; Dutting et al., 2021; Dütting et al., 2022; Castiglioni et al., 2022; 2023b).

**Learning in principal-agent problems**  The work that is most related to ours is (Zhu et al., 2023), which investigate a setting very similar to ours, though from an online learning perspective (see also our Section 5). Zhu et al. (2023) study general hidden-action principal-agent problem instances in which the principal faces multiple agent's types. They provide a regret bound of the order of $\widetilde{\mathcal{O}}(\sqrt{m} \cdot T^{1-1/(2m+1)})$ when the principal is restricted to contracts in $[0, 1]^m$, where $m$ is the number of outcomes. They also show that their regret bound can be improved to $\widetilde{\mathcal{O}}(T^{1-1/(m+2)})$ by making additional structural assumptions on the problem instances, including the *first-order stochastic dominance* (FOSD) condition. Moreover, Zhu et al. (2023) provide an (almost-matching) lower bound of $\Omega(T^{1-1/(m+2)})$ that holds even with a single agent's type, thus showing that the dependence on the number of outcomes $m$ at the exponent of $T$ is necessary in their setting. Notice that the regret bound by Zhu et al. (2023) is "close" to a linear dependence on $T$, even when $m$ is very small. In contrast, in Section 5 we show how our algorithm can be exploited to achieve a regret bound whose dependence on $T$ is of the order of $\widetilde{\mathcal{O}}(T^{4/5})$ (independent of $m$), when the number of agent's actions $n$ is constant. Another work that is closely related to ours is the one by Ho et al. (2014), who focus on designing a no-regret algorithm for a particular repeated principal-agent problem. However, their approach relies heavily on stringent structural assumptions, such as the FOSD condition. Finally, Cohen et al. (2022) study a repeated principal-agent interaction with a *risk-averse* agent, providing a no-regret algorithm that relies on the FOSD condition too.

**Learning in Stackelberg games**  The learning problem faced in this paper is closely related to the problem of learning an optimal strategy to commit to in Stackelberg games, where the leader repeatedly interacts with a follower by observing the follower's best-response action at each iteration. Letchford et al. (2009) propose an algorithm for the problem requiring the leader to randomly sample a set of available strategies in order to determine agent's best-response regions. The performances of such an algorithm depend on the volume of the smallest best-response region, and this considerably limits its generality. Peng et al. (2019) build upon the work by Letchford et al. (2009) by proposing an algorithm with more robust performances, being them independent of the volume of the smallest best-response region. The algorithm proposed in this paper borrows some ideas from that of Peng et al. (2019). However, it requires considerable additional machinery to circumvent the challenge posed by the fact that the principal does *not* observe agent's best-response actions, but only outcomes randomly sampled according to them. Other related works in the Stackelberg setting are (Bai et al., 2021), which proposes a model where both the leader and the follower learn through repeated interaction, and (Lauffer et al., 2023), which considers a scenario where the follower's utility is unknown to the leader, but it can be linearly parametrized.

**Assumptions relaxed compared to Stackelberg games**    In our work, we relax several limiting assumptions made in repeated Stackelber games (see, *e.g.*, (Letchford et al., 2009; Peng et al., 2019)) to learn an optimal commitment. Specifically, in repeated Stackelberg games either the best response regions have at least a constant volume or they are empty. Thanks to Lemma 8, we effectively address this limitation, showing that even when an optimal contract belongs to a zero-measured best-response region, we can still compute an approximately optimal solution. Furthermore, in Stackelberg games it is assumed that in cases where there are multiple best responses for the follower, any of them can be arbitrarily chosen. In contrast, we assume that the agent always breaks ties in favor of the leader as it is customary in the Stackelberg literature. Finally, our approach does not require the knowledge of the number of actions of the agent, differently from previously proposed algorithms.

# B    DETAILS ABOUT THE FIND-HS PROCEDURE

In this section, we describe in details the `Find-HS` procedure (Algorithm 6). The procedure takes as inputs a meta-action $d_i \in \mathcal{D}$ and a contract $p \in \mathcal{P}$, and it tries to find one of the approximate halfspaces defining a suitable approximate best-response region for the meta-action $d_i$. It may terminate either with a pair $(\widetilde{H}, d_k)$ such that $\widetilde{H}$ is an approximate halfspace in which $d_i$ is "supposedly better" than the meta-action $d_k \in \mathcal{D}$ or with a pair $(\varnothing, \bot)$, whenever a call to `Action-Oracle` returned $\bot$. Let us recall that, for ease of presentation, we assume that Algorithm 6 has access to all the variables declared in the `Try-Cover` procedure (Algorithm 3), namely $\mathcal{L}_d, \mathcal{U}_d, \mathcal{D}_d, \widetilde{H}_{ij}$, and $\Delta \widetilde{c}_{ij}$.

---

**Algorithm 6** Find-HS

---

**Require:** $d_i, p$
1: $p^1 \leftarrow$ any contract in $\mathcal{L}_{d_i}$
2: $p^2 \leftarrow p$
3: $d_j \leftarrow$ `Action-Oracle`$(p^1)$
4: $d_k \leftarrow$ `Action-Oracle`$(p^2)$
5: $y \leftarrow 18B\epsilon mn^2 + 2n\eta\sqrt{m}$
6: **while** $\|p^1 - p^2\|_2 > \eta$ **do**
7:     $p'_\omega \leftarrow (p^1_\omega + p^2_\omega)/2$ for all $\omega \in \Omega$        ▷ Middle point of the current line segment
8:     $d \leftarrow$ `Action-Oracle`$(p')$
9:     **if** $d = \bot$ **then return** $(\varnothing, \bot)$        ▷ Force termination of `Try-Cover`
10:     **if** $\mathcal{L}_d = \varnothing$ **then**        ▷ Add $d$ to to-be-processed meta-actions in `Try-Cover`
11:         $\mathcal{L}_d \leftarrow \{p'\}, \mathcal{C} \leftarrow \mathcal{C} \cup \{d\}$
12:     **end if**
13:     **if** $d \in \mathcal{D}_{d_i}$ **then**
14:         $p^1_\omega \leftarrow p'_\omega$ for all $\omega \in \Omega$
15:         $d_j \leftarrow d$
16:     **else**
17:         $p^2_\omega \leftarrow p'_\omega$ for all $\omega \in \Omega$
18:         $d_k \leftarrow d$
19:     **end if**
20: **end while**
21: $p'_\omega \leftarrow (p^1_\omega + p^2_\omega)/2$ for all $\omega \in \Omega$
22: **if** $d_j = d_i$ **then**        ▷ The approximate halfspace $\widetilde{H}_{ik}$ is the first one for $\mathcal{L}_{d_i}$
23:     $\Delta \widetilde{c}_{ik} \leftarrow \sum\limits_{\omega \in \Omega} \left( \widetilde{F}_{d_j,\omega} - \widetilde{F}_{d_k,\omega} \right) p'_\omega$
24: **else**        ▷ The approximate halfspace $\widetilde{H}_{ij}$ has already been computed
25:     $\Delta \widetilde{c}_{ik} \leftarrow \Delta \widetilde{c}_{ij} + \sum\limits_{\omega \in \Omega} \left( \widetilde{F}_{d_j,\omega} - \widetilde{F}_{d_k,\omega} \right) p'_\omega$
26: **end if**
27: $\widetilde{H} := \left\{ p \in \mathbb{R}^m_+ \mid \sum\limits_{\omega \in \Omega} \left( \widetilde{F}_{d_i,\omega} - \widetilde{F}_{d_k,\omega} \right) p_\omega \geq \Delta \widetilde{c}_{ik} - y \right\}$
28: **return** $(\widetilde{H}, d_k)$

---

Algorithm 6 works by performing a binary search on the line segment connecting $p$ with any contract in the (current) lower bound $\mathcal{L}_{d_i}$ of $d_i$ (computed by the `Try-Cover` procedure). At each iteration of the search, by letting $p^1, p^2 \in \mathcal{P}$ be the two extremes of the (current) line segment, the algorithm calls the `Action-Oracle` procedure on the contract $p' \in \mathcal{P}$ defined as the middle point of the segment. Then, if the procedure returns a meta-action $d \in \mathcal{D}$ that belongs to $\mathcal{D}_{d_i}$ (meaning that the approximate halfspace $\widetilde{H}_{ij}$ has already been computed), the algorithm sets $p^1$ to $p'$, while it sets $p^2$ to $p'$ otherwise. The binary search terminates when the length of the line segment is below a suitable threshold $\eta \geq 0$.

After the search has terminated, the algorithm computes an approximate halfspace. First, the algorithm computes the estimate $\Delta \widetilde{c}_{ik}$ of the cost difference between the meta-actions $d_i$ and $d_k$, where the latter is the last meta-action returned by `Action-Oracle` that does *not* belong to the set $\mathcal{D}_{d_i}$. Such an estimate is computed by using the two empirical distributions $\widetilde{F}_{d_j}$ and $\widetilde{F}_{d_k}$ that the `Action-Oracle` procedure associated with the meta-actions $d_j \in \mathcal{D}_{d_i}$ and $d_k$, respectively. In particular, $\Delta \widetilde{c}_{ik}$ is computed as the sum of $\Delta \widetilde{c}_{ij}$—the estimate of the cost difference between meta-actions $d_i$ and $d_j$ that has been computed during a previous call to Algorithm 6—and an estimate of the cost difference between the meta-actions $d_j$ and $d_k$, which can be computed by using the middle point of the line segment found by the binary search procedure (see Lines 23 and 25). Then, the desired approximate halfspace is the one defined by the hyperplane passing through the middle point $p' \in \mathcal{P}$ of the line segment computed by the binary search with coefficients given by the $(m+1)$-dimensional vector $[\widetilde{F}_{d_i}^\top - \widetilde{F}_{d_k}^\top, \Delta \widetilde{c}_{ik} - y\,]$ (see Line 27), where $y \geq 0$ is a suitably-defined value chosen so as to ensure that the approximate best-response regions satisfy the desired properties (see Lemma 10).

Notice that Algorithm 6 also needs some additional elements in order to ensure that the (calling) `Try-Cover` procedure is properly working. In particular, any time the `Action-Oracle` procedure returns $\bot$, Algorithm 6 terminates with $(\varnothing, \bot)$ as a return value (Line 9). This is needed to force the termination of the `Try-Cover` procedure (see Line 22 in Algorithm 3). Moreover, any time the `Action-Oracle` procedure returns a meta-action $d \notin \mathcal{C}$ such that $\mathcal{L}_d = \varnothing$, Algorithm 6 adds such a meta-action to $\mathcal{C}$ and it initializes its lower bound to the singleton $\{p'\}$, where $p'$ is the contract given as input to `Action-Oracle` (see Lines 10–11). This is needed to ensure that all the meta-actions in $\mathcal{D}$ are eventually considered by the (calling) `Try-Cover` procedure.

Before proving the main properties of Algorithm 6, we introduce the following useful definition:

**Definition 5.** *Given $y \geq 0$, a set of meta-actions $\mathcal{D}$ and a dictionary $\mathcal{F}$ of empirical distributions computed by means of Algorithm 2, for every pair of meta-actions $d_i, d_j \in \mathcal{D}$, we let $H_{ij}^y \subseteq \mathcal{P}$ be the set of contracts defined as follows:*

$$H_{ij}^y := \left\{ p \in \mathcal{P} \mid \sum_{\omega \in \Omega} \widetilde{F}_{d_i, \omega} p_\omega - c(d_i) \geq \sum_{\omega \in \Omega} \widetilde{F}_{d_j, \omega} p_\omega - c(d_j) - y \right\}.$$

*Furthermore, we let $H_{ij} := H_{ij}^0$.*

Next, we prove some technical lemmas related to Algorithm 6. Lemma 9 provides a bound on the gap between the cost difference $\Delta \widetilde{c}_{ik}$ estimated by Algorithm 6 and the "true" cost difference between the meta-actions $d_i$ and $d_k$ (see Defintion 4). Lemma 10 shows that the approximate halfspace computed by the algorithm is always included in the halfspace introduced in Definition 5 for a suitably-defined value of the parameter $y \geq 0$. Finally, Lemma 11 provides a bound on the number of rounds required by the algorithm in order to terminate.

**Lemma 9.** *Under the event $\mathcal{E}_\epsilon$, Algorithm 6 satisfies $|\Delta \widetilde{c}_{ik} - c(d_i) + c(d_k)| \leq 18B\epsilon m n^2 + 2n\eta \sqrt{m}$.*

*Proof.* We start by proving that, during any execution of Algorithm 6, the following holds:

$$\left| \sum_{\omega \in \Omega} \left( \widetilde{F}_{d_j, \omega} - \widetilde{F}_{d_k, \omega} \right) p'_\omega - c(d_j) + c(d_k) \right| \leq 18B\epsilon m n + 2\eta \sqrt{m}, \tag{1}$$

where $d_j, d_k \in \mathcal{D}$ are the meta-actions resulting from the binary search procedure and $p' \in \mathcal{P}$ is the middle point point of the segment after the binary search stopped. In order to prove Equation 1, we first let $a^1 := a^\star(p^1)$ and $a^2 := a^\star(p^2)$, where $p^1, p^2 \in \mathcal{P}$ are the two extremes of the line segment

resulting from the binary search procedure. By Lemma 1, under the event $\mathcal{E}_\epsilon$, it holds that $a^1 \in A(d_j)$ and $a^2 \in A(d_k)$ by definition. Moreover, we let $p^\star \in \mathcal{P}$ be the contract that belongs to the line segment connecting $p^1$ to $p^2$ and such that the agent is indifferent between actions $a^1$ and $a^2$, $i.e.$,

$$\sum_{\omega \in \Omega} \left( F_{a^1,\omega} - F_{a^2,\omega} \right) p^\star_\omega = c_{a^1} - c_{a^2}.$$

Then, we can prove the following:

$$\left| \sum_{\omega \in \Omega} \left( \widetilde{F}_{d_j,\omega} - \widetilde{F}_{d_k,\omega} \right) p'_\omega - c(d_j) + c(d_k) \right|$$

$$= \left| \sum_{\omega \in \Omega} \left( \widetilde{F}_{d_j,\omega} - \widetilde{F}_{d_k,\omega} \right) p'_\omega - c(d_j) + c_{a^1} - c_{a^1} + c_{a^2} - c_{a^2} + c(d_k) \right|$$

$$\leq \left| \sum_{\omega \in \Omega} \left( \widetilde{F}_{d_j,\omega} - \widetilde{F}_{d_k,\omega} \right) p'_\omega - c_{a^1} + c_{a^2} \right| + 8B\epsilon mn$$

$$= \left| \sum_{\omega \in \Omega} \left( \widetilde{F}_{d_j,\omega} - \widetilde{F}_{d_k,\omega} \right) p'_\omega - \sum_{\omega \in \Omega} \left( F_{a^1,\omega} - F_{a^2,\omega} \right) p^*_\omega \right| + 8B\epsilon mn$$

$$= \left| \sum_{\omega \in \Omega} \left( \widetilde{F}_{d_j,\omega} \, p'_\omega + F_{a^1,\omega} \, p'_\omega - F_{a^1,\omega} \, p'_\omega - F_{a^1,\omega} \, p^*_\omega \right) \right.$$

$$\left. + \sum_{\omega \in \Omega} \left( \widetilde{F}_{d_k,\omega} \, p'_\omega + F_{a^2,\omega} \, p'_\omega - F_{a^2,\omega} \, p'_\omega - F_{a^2,\omega} \, p^*_\omega \right) \right| + 8B\epsilon mn$$

$$\leq \| \widetilde{F}_{d_j} - F_{a^1} \|_\infty \| p' \|_1 + \| \widetilde{F}_{d_k} - F_{a^2} \|_\infty \| p' \|_1 + \left( \| F_{a^1} \|_2 + \| F_{a^2} \|_2 \right) \| p' - p^* \|_2 + 8B\epsilon mn$$

$$\leq 18B\epsilon mn + 2\eta\sqrt{m}.$$

The first inequality above is a direct consequence of Lemma 3 and an application of the triangular inequality, since $a^1 \in A(d_j)$ and $a^2 \in A(d_k)$ under the event $\mathcal{E}_\epsilon$. The second inequality follows by employing Holder's inequality. The third inequality holds by employing Lemma 1 and Definition 3, by observing that $\|p\|_1 \leq Bm$ and $\|F_a\| \leq \sqrt{m}$ for all $a \in \mathcal{A}$. Moreover, due to the definitions of $p'$ and $p^*$, and given how the binary search performed by Algorithm 6 works, it is guaranteed that $\|p' - p^*\|_2 \leq \eta$.

Next, we prove that, after any call to Algorithm 6, the following holds:

$$|\Delta\widetilde{c}_{ik} - c(c_i) + c(d_k)| \leq |\mathcal{D}_{d_i}| \left( 18B\epsilon mn + 2\eta\sqrt{m} \right),$$

where $d_i, d_k \in \mathcal{D}$ are the meta-actions defined by the binary search procedure in Algorithm 6. We prove the statement by induction on the calls to Algorithm 6 with the meta-action $d_i$ as input. The base case is the first time Algorithm 6 is called with $d_i$ as input. In that case, the statement is trivially satisfied by Equation 1 and the fact that $\mathcal{D}_{d_i} = \{d_i\}$. Let us now consider a generic call to Algorithm 6 with the meta-action $d_i$ as input. Then, we can write the following:

$$|\Delta\widetilde{c}_{ik} - c(c_i) + c(d_k)| = \left| \Delta\widetilde{c}_{ij} + \sum_{\omega \in \Omega} \left( \widetilde{F}_{d_j,\omega} - \widetilde{F}_{d_k,\omega} \right) p'_\omega - c(d_i) + c(d_k) \right|$$

$$\leq |\Delta\widetilde{c}_{ij} - c(d_i) + c(d_j)| + \left| \sum_{\omega \in \Omega} \left( \widetilde{F}_{d_j,\omega} - \widetilde{F}_{d_k,\omega} \right) p'_\omega - c(d_j) + c(d_k) \right|$$

$$\leq \left( |\mathcal{D}_{d_i}| - 1 \right) \left( 18B\epsilon mn + 2\eta\sqrt{m} \right) + 18B\epsilon mn + 2\eta\sqrt{m}$$

$$= |\mathcal{D}_{d_i}| \left( 18B\epsilon mn + 2\eta\sqrt{m} \right),$$

where the first inequality holds by applying the triangular inequality and the second one by using the inductive hypothesis. $\qquad \square$

**Lemma 10.** *Let $(\widetilde{H}, d_k)$ be the pair returned by Algorithm 6 when given as inputs $\mathcal{L}_{d_i}$ and $p \in \mathcal{P}$ for some meta-action $d_i \in \mathcal{D}$. Then, under the event $\mathcal{E}_\epsilon$, it holds:*

$$H_{ik} \subseteq \widetilde{H} \subseteq H_{ik}^y,$$

*with $y = 18B\epsilon mn^2 + 2n\eta\sqrt{m}$.*

*Proof.* We start by showing that $H_{ik}$ is a subset of $\widetilde{H}_{ik}$. Let $p$ be a contract belonging to $H_{ik}$, then according to Definition 5 the following inequality holds:

$$\sum_{\omega \in \Omega} \widetilde{F}_{d_i,\omega} p_\omega - \sum_{\omega \in \Omega} \widetilde{F}_{d_k,\omega} p_\omega \geq \Delta \widetilde{c}_{ik} - y,$$

with $y = 18B\epsilon mn^2 + 2n\eta\sqrt{m}$ as prescribed by Lemma 9. This shows that $p \in \widetilde{H}_{ik}$, according to Definition 5. We now consider the case in which $p \in \widetilde{H}_{ik}$. Using the definition of $\widetilde{H}_{ik}$ and Lemma 9, we have:

$$\sum_{\omega \in \Omega} \widetilde{F}_{d_i,\omega} p_\omega - \sum_{\omega \in \Omega} \widetilde{F}_{d_k,\omega} p_\omega = \Delta \widetilde{c}_{ik} \geq c(d_i) - c(d_k) - y,$$

showing that $p \in H_{ik}^y$ with $y = 18B\epsilon mn^2 + 2n\eta\sqrt{m}$. $\qquad\square$

**Lemma 11.** *Under the event $\mathcal{E}_\epsilon$, the number of rounds required by Algorithm 6 to terminate with an approximate separating hyperplane is at most $\mathcal{O}\left(q \log\left(^{Bm}/_\eta\right)\right)$.*

*Proof.* The lemma can be proven by observing that the number of rounds required by the binary search performed in Line 6 of Algorithm 6 is at most $\mathcal{O}(\log(^D/_\eta))$, where $D$ represents the distance over which the binary search is performed. In our case, we have $D \leq \sqrt{2}Bm$, which represents the maximum distance between two contracts in $\mathcal{P}$. Additionally, noticing that we invoke the `Action-Oracle` algorithm at each iteration, the total number of rounds required by `Find-HS` is $\mathcal{O}(q \log(\frac{\sqrt{2}Bm}{\eta}))$, as the number of rounds required by `Action-Oracle` is bounded by $q$. $\qquad\square$

## C  ADDITIONAL DETAILS ABOUT THE DISCOVER-AND-COVER ALGORITHM

In this section, we provide all the details about the `Discover-and-Cover` (Algorithm 1) algorithm that are omitted from the main body of the paper. In particular:

- Appendix C.1 gives a summary of the definitions of the parameters required by Algorithm 1, which are set as needed in the proofs provided in the rest of this section.
- Appendix C.2 provides the proofs of the lemmas related to the `Action-Oracle` procedure.
- Appendix C.3 provides the proofs of the lemmas related to the `Try-Cover` procedure.
- Appendix C.4 provides the proofs of the lemmas related to the `Find-Contract` procedure.
- Appendix C.5 provides the proof of the final result related to Algorithm 1 (Theorem 2).

### C.1  DEFINITIONS OF THE PARAMETERS IN ALGORITHM 1

The parameters required by Algorithm 1 are defined as follows:

- $\epsilon := \dfrac{\rho^2}{32^2 Bm^2 n^2}$

- $\eta := \dfrac{\epsilon\sqrt{m}n}{2}$

- $\alpha := \dfrac{\delta}{2n^3 \left[\log\left(\frac{2Bm}{\eta}\right) + \binom{m+n+1}{m}\right]}$

- $q := \left\lceil \dfrac{1}{2\epsilon^2} \log\left(\dfrac{2m}{\alpha}\right) \right\rceil$

## C.2 Proofs of the lemmas related to the Action-Oracle procedure

**Lemma 1.** *Given a set $\mathcal{D}$ of meta-actions and a dictionary $\mathcal{F}$ of empirical distributions computed by means of Algorithm 2, under the event $\mathcal{E}_\epsilon$, if Algorithm 2 returns a meta-action $d \in \mathcal{D}$ for a contract $p \in \mathcal{P}$ given as input, then it holds that $a^\star(p) \in A(d)$.*

*Proof.* For ease of presentation, in this proof we adopt the following additional notation. Given a set $\mathcal{D}$ of meta-actions and a dictionary $\mathcal{F}$ of empirical distributions computed by means of Algorithm 2, for every meta-action $d \in \mathcal{D}$, we let $\mathcal{P}_d \subseteq \mathcal{P}$ be the set of all the contracts that have been provided as input to Algorithm 2 during an execution in which it computed an empirical distribution that belongs to $\mathcal{F}[d]$. Moreover, we let $I(d) \subseteq \mathcal{A}$ be the set of all the agent's actions that have been played as a best response by the agent during at least one of such executions of Algorithm 2. Formally, by exploiting the definition of $\mathcal{P}_d$, we can write $I(d) := \bigcup_{p \in \mathcal{P}_d} a^\star(p)$.

First, we prove the following crucial property of Algorithm 2:

**Property 4.** *Given a set $\mathcal{D}$ of meta-actions and a dictionary $\mathcal{F}$ of empirical distributions computed by means of Algorithm 2, under the event $\mathcal{E}_\epsilon$, ti holds that $I(d) \cap I(d') = \varnothing$ for every pair of (different) meta-actions $d \neq d' \in \mathcal{D}$.*

In order to show that Property 4 holds, we assume by contradiction that there exist two different meta-actions $d \neq d' \in \mathcal{D}$ such that $a \in I(d)$ and $a \in I(d')$ for some agent's action $a \in \mathcal{A}$, namely that $I(d) \cap I(d') \neq \varnothing$. This implies that there exist two contracts $p \in \mathcal{P}_d$ and $p' \in \mathcal{P}_{d'}$ such that $a^\star(p) = a^\star(p') = a$. Let $\widetilde{F} \in \mathcal{F}[d]$ and $\widetilde{F}' \in \mathcal{F}[d']$ be the empirical distributions over outcomes computed by Algorithm 2 when given as inputs the contracts $p$ and $p'$, respectively. Then, under the event $\mathcal{E}_\epsilon$, we have that $\|\widetilde{F} - \widetilde{F}'\|_\infty \leq 2\epsilon$, since the two empirical distributions are generated by sampling from the same (true) distribution $F_a$. Clearly, the way in which Algorithm 2 works implies that such empirical distributions are associated with the same meta-action and put in the same entry of the dictionary $\mathcal{F}$. This contradicts the assumption that $d \neq d'$, proving that Property 4 holds.

Next, we show the following additional crucial property of Algorithm 2:

**Property 5.** *Let $\mathcal{D}$ be a set of meta-actions and $\mathcal{F}$ be a dictionary of empirical distributions computed by means of Algorithm 2. Suppose that the following holds for every $d \in \mathcal{D}$ and $a, a' \in I(d)$:*

$$\|F_a - F_{a'}\|_\infty \leq 4\epsilon(|I(d)| - 1). \tag{2}$$

*Then, if Algorithm 2 is run again, under the event $\mathcal{E}_\epsilon$ the same condition continues to hold for the set of meta-actions and the dictionary obtained after the execution of Algorithm 2.*

For ease of presentation, in the following we call $\mathcal{D}^{\text{old}}$ and $\mathcal{F}^{\text{old}}$ the set of meta-actions and the dictionary of empirical distributions, respectively, before the last execution of Algorithm 2, while we call $\mathcal{D}^{\text{new}}$ and $\mathcal{F}^{\text{new}}$ those obtained after the last run of Algorithm 2. Moreover, in order to avoid any confusion, given $\mathcal{D}^{\text{old}}$ and $\mathcal{F}^{\text{old}}$, we write $I^{\text{old}}(d)$ in place of $I(d)$ for any meta-action $d \in \mathcal{D}^{\text{old}}$. Similarly, given $\mathcal{D}^{\text{new}}$ and $\mathcal{F}^{\text{new}}$, we write $I^{\text{new}}(d)$ in place of $I(d)$ for any $d \in \mathcal{D}^{\text{new}}$.

In order to show that Property 5 holds, let $p \in \mathcal{P}$ be the contract given as input to Algorithm 2 during its last run. Next, we prove that, no matter how Algorithm 2 updates $\mathcal{D}^{\text{old}}$ and $\mathcal{F}^{\text{old}}$ in order to obtain $\mathcal{D}^{\text{new}}$ and $\mathcal{F}^{\text{new}}$, the condition in Property 5 continues to hold. We split the proof in three cases.

1. If $|\mathcal{D}^\diamond| = 0$, then $\mathcal{D}^{\text{new}} = \mathcal{D}^{\text{old}} \cup \{d^\diamond\}$, where $d^\diamond$ is a new meta-action. Since $\mathcal{F}^{\text{new}}$ is built by adding a new entry $\mathcal{F}^{\text{new}}[d^\diamond] = \{\widetilde{F}\}$ to $\mathcal{F}^{\text{old}}$ while leaving all the other entries unchanged, it holds that $I^{\text{new}}(d^\diamond) = \{a^\star(p)\}$ and $I^{\text{new}}(d) = I^{\text{old}}(d)$ for all $d \in \mathcal{D}^{\text{old}}$. As a result, the condition in Equation 2 continues to hold for all the meta-actions $d \in \mathcal{D}^{\text{new}} \setminus \{d^\diamond\}$. Moreover, for the meta-action $d^\diamond$, the following holds:

$$0 = \|F_{a^\star(p)} - F_{a^\star(p)}\|_\infty \leq 4\epsilon(|I^{\text{new}}(d^\diamond)| - 1) = 0,$$

as $|I^{\text{new}}(d^\diamond)| = 1$. This proves that the condition in Equation 2 also holds for $d^\diamond$.

2. If $|\mathcal{D}^\diamond| = 1$, then $\mathcal{D}^{\text{new}} = \mathcal{D}^{\text{old}}$. We distinguish between two cases.

(a) In the case in which $a^\star(p) \in \bigcup_{d \in \mathcal{D}^{\mathrm{old}}} I^{\mathrm{old}}(d)$, Property 4 immediately implies that $I^{\mathrm{new}}(d) = I^{\mathrm{old}}(d)$ for all $d \in \mathcal{D}^{\mathrm{new}} = \mathcal{D}^{\mathrm{old}}$. Indeed, if this is *not* the case, then there would be two different meta-actions $d \neq d' \in \mathcal{D}^{\mathrm{new}} = \mathcal{D}^{\mathrm{old}}$ such that $I^{\mathrm{new}}(d) = I^{\mathrm{old}}(d) \cup \{a^\star(p)\}$ (since $\mathcal{F}^{\mathrm{new}}[d] = \mathcal{F}^{\mathrm{old}}[d] \cup \{\widetilde{F}\}$) and $a^\star(p) \in I^{\mathrm{new}}(d') = I^{\mathrm{old}}(d')$, contradicting Property 4. As a result, the condition in Equation 2 continues to holds for all the meta-actions after the execution of Algorithm 2.

(b) In the case in which $a^\star(p) \notin \bigcup_{d \in \mathcal{D}^{\mathrm{old}}} I^{\mathrm{old}}(d)$, the proof is more involved. Let $d \in \mathcal{D}^{\mathrm{new}} = \mathcal{D}^{\mathrm{old}}$ be the (unique) meta-action in the set $\mathcal{D}^\diamond$ computed by Algorithm 2. Notice that $I^{\mathrm{new}}(d) = I^{\mathrm{old}}(d) \cup \{a^\star(p)\}$ by how Algorithm 2 works. As a first step, we show that, for every pair of actions $a, a' \in I^{\mathrm{new}}(d)$ with $a = a^\star(p)$, Equation 2 holds. Under the event $\mathcal{E}_\epsilon$, it holds that $\|F_{a^\star(p)} - \widetilde{F}\|_\infty \leq \epsilon$, where $\widetilde{F}$ is the empirical distribution computed by Algorithm 2. Moreover, since the meta-action $d$ has been added to $\mathcal{D}^\diamond$ by Algorithm 2, there exists an empirical distribution $F \in \mathcal{F}[d]$ such that $\|\widetilde{F} - F\|_\infty \leq 2\epsilon$, and, under the event $\mathcal{E}_\epsilon$, there exists an action $a'' \in I^{\mathrm{old}}(d)$ such that $\|F - F_{a''}\|_\infty \leq \epsilon$. Then, by applying the triangular inequality we can show that:

$$\|F_{a^\star(p)} - F_{a''}\|_\infty \leq \|F_{a^\star(p)} - \widetilde{F}\|_\infty + \|\widetilde{F} - F\|_\infty + \|F - F_{a''}\|_\infty \leq 4\epsilon.$$

By using the fact that the condition in Equation 2 holds for $\mathcal{D}^{\mathrm{new}}$ and $\mathcal{F}^{\mathrm{new}}$, for every action $a' \in I^{\mathrm{new}}(d)$ it holds that:

$$\begin{aligned}
\|F_{a^\star(p)} - F_{a'}\|_\infty &\leq \|F_{a^\star(p)} - F_{a''}\|_\infty + \|F_{a''} - F_{a'}\|_\infty \\
&\leq 4\epsilon(|I^{\mathrm{old}}(d)| - 1) + 4\epsilon \\
&\leq 4\epsilon|I^{\mathrm{old}}(d)| \\
&= 4\epsilon(|I^{\mathrm{new}}(d)| - 1),
\end{aligned}$$

where the last equality holds since $|I^{\mathrm{new}}(d)| = |I^{\mathrm{old}}(d)| + 1$, as $a^\star(p) \notin \bigcup_{d \in \mathcal{D}^{\mathrm{old}}} I^{\mathrm{old}}(d)$. This proves that Equation 2 holds for every pair of actions $a, a' \in I^{\mathrm{new}}(d)$ with $a = a^\star(p)$. For all the other pairs of actions $a, a' \in I^{\mathrm{new}}(d)$, the equation holds since it was already satisfied before the last execution of Algorithm 2. Analogously, given that $I^{\mathrm{new}}(d) = I^{\mathrm{old}}(d)$ for all the meta-actions in $\mathcal{D}^{\mathrm{new}} \setminus \{d\}$, we can conclude that the condition in Equation 2 continues to holds for such meta-actions as well.

3. If $|\mathcal{D}^\diamond| > 1$, then $\mathcal{D}^{\mathrm{new}} = (\mathcal{D}^{\mathrm{old}} \setminus \mathcal{D}^\diamond) \cup \{d^\diamond\}$, where $d^\diamond$ is a new meta-action. Clearly, the condition in Equation 2 continues to hold for all the meta-actions in $\mathcal{D}^{\mathrm{old}} \setminus \mathcal{D}^\diamond$. We only need to show that the condition holds for $d^\diamond$. We distinguish between two cases.

(a) In the case in which $a^\star(p) \in \bigcup_{d \in \mathcal{D}^{\mathrm{old}}} I^{\mathrm{old}}(d)$, let us first notice that $a^\star(p) \in I^{\mathrm{old}}(d)$ for some $d \in \mathcal{D}^\diamond$, otherwise Property 4 would be violated (as $a^\star(p) \in I^{\mathrm{new}}(d^\diamond)$ by definition). Moreover, it is easy to see that $I^{\mathrm{new}}(d^\diamond) = \bigcup_{d \in \mathcal{D}^\diamond} I^{\mathrm{old}}(d)$ and, additionally, $I^{\mathrm{old}}(d) \cap I^{\mathrm{old}}(d') = \varnothing$ for all $d \neq d' \in \mathcal{D}^\diamond$, given how Algorithm 2 works and thanks to Property 4. In the following, for ease of presentation, we assume w.l.o.g. that the meta-actions in $\mathcal{D}^\diamond$ are indexed by natural numbers so that $\mathcal{D}^\diamond := \{d_1, \ldots d_{|\mathcal{D}^\diamond|}\}$ and $a^\star(p) \in I^{\mathrm{old}}(d_1)$. Next, we show that Equation 2 holds for every pair $a, a' \in I^{\mathrm{new}}(d^\diamond)$. First, by employing an argument similar to the one used to prove Point 2.2, we can show that, for every $d_j \in \mathcal{D}^\diamond$ with $j > 1$, there exists an action $a'' \in I^{\mathrm{old}}(d_j)$ such that $\|F_{a^\star(p)} - F_{a''}\|_\infty \leq 4\epsilon$. Then, for every pair of actions $a, a' \in I^{\mathrm{new}}(d^\diamond)$ such that $a \in I^{\mathrm{old}}(d_1)$ and $a' \in I^{\mathrm{old}}(d_j)$ for some $d_j \in \mathcal{D}^\diamond$ with $j > 1$, the following holds:

$$\begin{aligned}
\|F_a - F_{a'}\|_\infty &\leq \|F_a - F_{a^\star(p)}\|_\infty + \|F_{a^\star(p)} - F_{a''}\|_\infty + \|F_{a''} - F_{a'}\|_\infty \\
&\leq 4\epsilon(|I^{\mathrm{old}}(d_1)| - 1) + 4\epsilon + 4\epsilon(|I^{\mathrm{old}}(d_j)| - 1) \\
&= 4\epsilon(|I^{\mathrm{old}}(d_1)| + |I^{\mathrm{old}}(d_j)| - 1) \\
&\leq 4\epsilon(|I^{\mathrm{new}}(d^\diamond)| - 1),
\end{aligned}$$

where the first inequality follows from an application of the triangular inequality, the second one from the fact that Equation 2 holds before the last execution of Algorithm 2, while the last inequality holds since $|I^{\mathrm{new}}(d^\diamond)| = \sum_{d \in \mathcal{D}^\diamond} |I^{\mathrm{old}}(d)|$ given that $I^{\mathrm{old}}(d) \cap$

$I^{\text{old}}(d') = \varnothing$ for all $d \neq d' \in \mathcal{D}^\diamond$ thanks to Property 4. As a final step, we show that Equation 2 holds for every pair of actions $a, a' \in I^{\text{new}}(d^\diamond)$ such that $a \in I^{\text{old}}(d_i)$ and $a' \in I^{\text{old}}(d_j)$ for some $d_i, d_j \in \mathcal{D}^\diamond$ with $i \neq j > 1$. By the fact that $d_i, d_j \in \mathcal{D}^\diamond$ and triangular inequality, it follows that there exist $a'' \in I^{\text{old}}(d_i)$ and $a''' \in I^{\text{old}}(d_j)$ such that $\|F_{a''} - \widetilde{F}\|_\infty \leq 3\epsilon$ and $\|F_{a'''} - \widetilde{F}\|_\infty \leq 3\epsilon$ under $\mathcal{E}_\epsilon$. Then, with steps similar to those undertaken above, we can prove the following:

$$\|F_a - F_{a'}\|_\infty \leq \|F_a - F_{a''}\|_\infty + \|F_{a''} - \widetilde{F}\|_\infty + \|\widetilde{F} - F_{a'''}\|_\infty + \|F_{a'''} - F_{a'}\|_\infty$$
$$\leq 4\epsilon(|I^{\text{old}}(d_i)| - 1) + 3\epsilon + 3\epsilon + 4\epsilon(|I^{\text{old}}(d_j)| - 1)$$
$$= 4\epsilon(|I^{\text{old}}(d_i)| + |I^{\text{old}}(d_j)| - 1) + 2\epsilon$$
$$\leq 4\epsilon(|I^{\text{new}}(d^\diamond)| - 1),$$

which shows that Equation 2 is satisfied for all the pairs of actions $a, a' \in I^{\text{new}}(d^\diamond)$.

(b) In the case in which $a^\star(p) \notin \bigcup_{d \in \mathcal{D}^{\text{old}}} I^{\text{old}}(d)$, let us first observe that $I^{\text{new}}(d^\diamond) = \bigcup_{d \in \mathcal{D}^\diamond} I^{\text{old}}(d) \cup \{a^\star(p)\}$ and $I^{\text{old}}(d) \cap I^{\text{old}}(d') = \varnothing$ for all $d \neq d' \in \mathcal{D}^\diamond$, given how Algorithm 2 works and thanks to Property 4. Next, we show that Equation 2 holds for every pair of actions $a, a' \in I^{\text{new}}(d^\diamond)$. As a first step, we consider the case in which $a = a^\star(p)$, and we show that Equation 2 holds for every $a' \in I^{\text{new}}(d^\diamond)$ such that $a' \in I^{\text{old}}(d)$ for some $d \in \mathcal{D}^\diamond$. In order to show this, we first observe that, given how Algorithm 2 works, there exists $F \in \mathcal{F}^{\text{old}}[d]$ such that $\|\widetilde{F} - F\| \leq 2\epsilon$, and, under the event $\mathcal{E}_\epsilon$, there exists an action $a'' \in I^{\text{old}}(d)$ such that $\|F_{a''} - F\| \leq \epsilon$. Then:

$$\|F_{a^\star(p)} - F_{a'}\|_\infty \leq \|F_{a^\star(p)} - \widetilde{F}\|_\infty + \|\widetilde{F} - F\|_\infty + \|F - F_{a''}\|_\infty + \|F_{a''} - F_{a'}\|_\infty$$
$$\leq 4\epsilon + 4\epsilon(|I^{\text{old}}(d)| - 1)$$
$$\leq 4\epsilon(|I^{\text{new}}(d^\diamond)| - 1),$$

where the last inequality holds since $|I_{d^\diamond}^{\text{new}}| = \sum_{d \in \mathcal{D}^\diamond} |I^{\text{old}}(d)|$ as $I^{\text{old}}(d) \cap I^{\text{old}}(d') = \varnothing$ for all $d \neq d' \in \mathcal{D}^\diamond$. As a second step, we consider the case in which $a \in I^{\text{old}}(d)$ and $a' \in I^{\text{old}}(d')$ for some pair $d \neq d' \in \mathcal{D}^\diamond$. Given how Algorithm 2 works, there exist $F \in \mathcal{F}^{\text{old}}[d]$ and $F' \in \mathcal{F}^{\text{old}}[d']$ such that $\|F - \widetilde{F}\| \leq 2\epsilon$ and $\|F' - \widetilde{F}\| \leq 2\epsilon$. Moreover, by the fact that $d, d' \in \mathcal{D}^\diamond$ and the triangular inequality, it follows that there exist $a'' \in I^{\text{old}}(d)$ and $a''' \in I^{\text{old}}(d')$ such that $\|F_{a''} - \widetilde{F}\|_\infty \leq 3\epsilon$ and $\|F_{a'''} - \widetilde{F}\|_\infty \leq 3\epsilon$ under the event $\mathcal{E}_\epsilon$. Then, the following holds:

$$\|F_a - F_{a'}\|_\infty \leq \|F_a - F_{a''}\|_\infty + \|F_{a''} - \widetilde{F}\|_\infty + \|\widetilde{F} - F_{a'''}\|_\infty + \|F_{a'''} - F_{a'}\|_\infty$$
$$\leq 4\epsilon(|I^{\text{old}}(d)| - 1) + 3\epsilon + 3\epsilon + 4\epsilon(|I^{\text{old}}(d')| - 1)$$
$$= 4\epsilon(|I^{\text{old}}(d)| + |I^{\text{old}}(d')| - 1) + 2\epsilon$$
$$\leq 4\epsilon(|I^{\text{new}}(d^\diamond)| - 1).$$

Finally, by observing that the size of $I(d)$ for each possible set of meta-actions $\mathcal{D}$ is bounded by the number of agent's actions $n$, for each $a, a' \in I(d)$ and $d \in \mathcal{D}$ we have that:

$$\|F_a - F_{a'}\| \leq 4n\epsilon. \tag{3}$$

Then, under the event $\mathcal{E}_\epsilon$, by letting $a' \in I(d)$ be the action leading to the empirical distribution $\widetilde{F}_d$ computed at Line 19 in Algorithm 2 we have $\|\widetilde{F}_d - F_{a'}\|_\infty \leq \epsilon$. Finally, combining the two inequalities, we get:

$$\|\widetilde{F}_d - F_{a^\star(p)}\|_\infty \leq \|\widetilde{F}_d - F_{a'}\|_\infty + \|F_{a'} - F_{a^\star(p)}\|_\infty \leq 4\epsilon n + \epsilon \leq 5\epsilon n.$$

This implies that $a^\star(p) \in A(d)$.

To conclude the proof we show that $I(d) \subseteq A(d)$. Let $a' \in I(d)$ be an arbitrary action, and let $\widetilde{F}_d$ be the empirical distribution computed by Algorithm 2 by sampling from $F_a$. Then for each $a'' \in I(d)$ we have:

$$\|\widetilde{F}_d - F_{a''}\|_\infty \leq \|\widetilde{F}_d - F_{a'}\|_\infty + \|F_{a'} - F_{a''}\|_\infty \leq 4\epsilon n + \epsilon \leq 5\epsilon n.$$

Since the latter argument holds for all the actions $a \in I(d)$ this hows that

$$\|F_a - F_{a'}\|_\infty \leq 4n\epsilon, \tag{4}$$

for all the actions $a, a' \in A(d)$. $\qquad\square$

**Lemma 2.** *Under the event $\mathcal{E}_\epsilon$, Algorithm 2 returns $\perp$ at most $2n$ times.*

*Proof.* As a first step, we observe that under the event $\mathcal{E}_\epsilon$, the size of $\mathcal{D}$ increases whenever the principal observes an empirical distribution $\widetilde{F}$ that satisfies $\|\widetilde{F} - \widetilde{F}'\|_\infty \geq 2\epsilon$ for every $\widetilde{F}' \in \mathcal{F}$. This condition holds when the agent chooses an action that has not been selected before. This is because, if the principal commits to a contract implementing an action the agent has already played, the resulting estimated empirical distribution $\widetilde{F}$ must satisfy $\|\widetilde{F} - F\|_\infty \leq 2\epsilon$ for some $d \in \mathcal{D}$ and $F \in \mathcal{F}_d$, as guaranteed by the event $\mathcal{E}_\epsilon$. Consequently, the cardinality of $\mathcal{D}$ can increase at most $n$ times, which corresponds to the total number of actions. Furthermore, we observe that the cardinality of $\mathcal{D}$ can decrease by merging one or more meta-actions into a single one, with the condition $|\mathcal{D}| \geq 1$. Therefore, in the worst case the cardinality of $\mathcal{D}$ is reduced by one $n$ times, resulting in $2n$ being the maximum number of times Algorithm 2 returns $\perp$. $\square$

**Lemma 3.** *Given a set $\mathcal{D}$ of meta-actions and a dictionary $\mathcal{F}$ of empirical distributions computed by means of Algorithm 2, under the event $\mathcal{E}_\epsilon$, for every meta-action $d \in \mathcal{D}$ and associated action $a \in A(d)$ it holds that $|c(d) - c_a| \leq 4B\epsilon mn$.*

*Proof.* Let $d \in \mathcal{D}$ be a meta-action and assume that the event $\mathcal{E}_\epsilon$ holds. Let $a' \in \arg\min_{a \in A(d)} c_a$ and let $p'$ be a contract such that $a^\star(p') = a'$. By employing Definition 3, we know that for any action $a \in A(d)$, there exists a contract $p \in \mathcal{P}$ such that $a^\star(p) = a$. Then the following inequalities hold:

$$4B\epsilon mn \geq \|F_a - F_{a'}\|_\infty \|p\|_1 \geq \sum_{\omega \in \Omega} (F_{a,\omega} - F_{a',\omega}) p_\omega \geq c_a - c_{a'}, \tag{5}$$

and, similarly,

$$4B\epsilon mn \geq \|F_{a'} - F_a\|_\infty \|p'\|_1 \geq \sum_{\omega \in \Omega} (F_{a',\omega} - F_{a,\omega}) p'_\omega \geq c_{a'} - c_a. \tag{6}$$

In particular, in the chains of inequalities above, the first '$\geq$' holds since $\|F_a - F_{a'}\|_\infty \leq 4\epsilon n$ by Lemma 1 and the fact that the norm $\|\cdot\|_1$ of contracts in $\mathcal{P}$ is upper bounded by $Bm$. Finally, by applying Equations 5 and 6, we have that:

$$|c_a - c_{a'}| = |c_a - c(d)| \leq 4B\epsilon mn,$$

for every possible action $a \in A(d)$ since $c_{a'} = c(d)$ by definition, concluding the proof. $\square$

**Lemma 4.** *Given a set $\mathcal{D}$ of meta-actions and a dictionary $\mathcal{F}$ of empirical distributions computed by means of Algorithm 2, under the event $\mathcal{E}_\epsilon$, for every meta-action $d \in \mathcal{D}$, associated action $a \in A(d)$, and contract $p \in \mathcal{P}$ it holds $|\sum_{\omega \in \Omega} \widetilde{F}_{d,\omega}\, p_\omega - c(d) - \sum_{\omega \in \Omega} F_{a,\omega}\, p_\omega + c_a| \leq 9B\epsilon mn$.*

*Proof.* To prove the lemma we observe that for each action $a \in A(d)$ it holds:

$$\left| \sum_{\omega \in \Omega} \widetilde{F}_{d,\omega} p_\omega - c(d) - \sum_{\omega \in \Omega} F_{a,\omega} p_\omega + c_a \right| \leq \|\widetilde{F}_d - F_a\|_\infty \|p\|_1 + |c(d) - c_a|$$

$$\leq 5B\epsilon mn + 4B\epsilon mn = 9B\epsilon mn,$$

where the first inequality holds by applying the triangular inequality and Holder's inequality. The second inequality holds by employing Lemma 3 and leveraging Definition 3 that guarantees $\|\widetilde{F}_d - F_a\|_\infty \leq 5\epsilon n$ for every action $a \in A(d)$. Additionally, we observe that the $\|\cdot\|_1$ norm of contracts in $\mathcal{P}$ is upper bounded by $Bm$, which concludes the proof. $\square$

We conclude the subsection by providing an auxiliary lemmas related to the `Action-Oracle` procedure that will be employed to bound the probability of the clean event $\mathcal{E}_\epsilon$ in the proof of Theorem 2.

**Lemma 12.** *Given $\alpha, \epsilon \in (0, 1)$, let $\widetilde{F} \in \Delta_\Omega$ be the empirical distribution computed by Algorithm 2 with $q := \lceil \frac{1}{2\epsilon^2} \log\left(\frac{2m}{\alpha}\right) \rceil$ for a contract $p \in \mathcal{P}$ given as input. Then, it holds that $\mathbb{E}[\widetilde{F}_\omega] = F_{a^\star(p),\omega}$ for all $\omega \in \Omega$ and, with probability of at least $1 - \alpha$, it also holds that $\|\widetilde{F} - F_{a^\star(p)}\|_\infty \leq \epsilon$.*

*Proof.* By construction, the empirical distribution $\widetilde{F} \in \Delta_\Omega$ is computed from $q$ i.i.d. samples drawn according to $F_{a^\star(p)}$, with each $\omega \in \Omega$ being drawn with probability $F_{a^\star(p),\omega}$. Therefore, the empirical distribution $\widetilde{F}$ is a random vector supported on $\Delta_\Omega$, whose expectation is such that $\mathbb{E}[\widetilde{F}_\omega] = F_{a^\star(p),\omega}$ for all $\omega \in \Omega$. Moreover, by Hoeffding's inequality, for every $\omega \in \Omega$, we have that:

$$\mathbb{P}\left\{ |\widetilde{F}_\omega - \mathbb{E}[\widetilde{F}_\omega]| \geq \epsilon \right\} = \mathbb{P}\left\{ |\widetilde{F}_\omega - F_{a^\star(p),\omega}| \geq \epsilon \right\} \geq 1 - 2e^{-2q\epsilon^2}. \tag{7}$$

Then, by employing a union bound and Equation 7 we have that:

$$\mathbb{P}\left\{ \|\widetilde{F} - F_{a^\star(p)}\|_\infty \leq \epsilon \right\} = \mathbb{P}\left\{ \bigcap_{\omega \in \Omega} \left\{ |\widetilde{F}_\omega - F_{a^\star(p),\omega}| \leq \epsilon \right\} \right\} \geq 1 - 2me^{-2q\epsilon^2} \geq 1 - \alpha,$$

where the last inequality holds by definition of $q$. $\qquad\square$

### C.3 PROOFS OF THE LEMMAS RELATED TO THE TRY-COVER PROCEDURE

To prove Lemma 5, we first introduce Definition 6 for a given a set of meta-actions $\mathcal{D}$. This definition associates to each meta-acxxtion $d$ the set of contracts in which the agent's utility, computed employing the empirical distribution over outcomes returned by the Action-Oracle procedure and the cost of a meta-action introduced in Definition 4, is greater or equal to the utility computed with the same quantities for all the remaining meta-actions in $\mathcal{D}$. Formally we have that:

**Definition 6.** *Given a set of meta-actions $\mathcal{D}$, we let $\mathcal{P}_{d_i}(\mathcal{D}) \subseteq \mathcal{P}$ be the set defined as follows:*

$$\mathcal{P}_{d_i}(\mathcal{D}) := \left\{ p \in \mathcal{P} \mid \sum_{\omega \in \Omega} \widetilde{F}_{d_i,\omega} p_\omega - c(d_i) \geq \sum_{\omega \in \Omega} \widetilde{F}_{d_j,\omega} p_\omega - c(d_j) \;\; \forall d_j \in \mathcal{D} \right\}.$$

It is important to notice that we can equivalently formulate Definition 6 by means of Definition 5. More specifically, given a set of met actions $\mathcal{D}$, for each $d_i \in \mathcal{D}$ we let $\mathcal{P}_{d_i}(\mathcal{D}) := \cap_{j \in \mathcal{D}} H_{ij}$ be the intersection of a subset of the halfspaces introduced in Definition 5.

As a second step, we introduce two useful lemmas. Lemma 13 shows that for any set of meta-actions $\mathcal{D}$, the union of the sets $\mathcal{P}_d(\mathcal{D})$ over all $d \in \mathcal{D}$ is equal to $\mathcal{P}$. On the other hand, Lemma 14 shows that the set $\mathcal{P}_d(\mathcal{D})$ is a subset of the upper bounds $\mathcal{U}_d$ computed by the Try-Cover procedure.

**Lemma 13.** *Given a set of meta-actions $\mathcal{D}$ it always holds $\cup_{d \in \mathcal{D}} \mathcal{P}_d(\mathcal{D}) = \mathcal{P}$.*

*Proof.* The lemma follows observing that, for each $p \in \mathcal{P}$, there always exits a $d_i \in \mathcal{D}$ such that:

$$\sum_{\omega \in \Omega} \widetilde{F}_{d_i,\omega} p_\omega - c(d_i) \geq \sum_{\omega \in \Omega} \widetilde{F}_{d_j,\omega} p_\omega - c(d_j) \;\; \forall d_j \in \mathcal{D}.$$

This is due to the fact that the cardinality of $\mathcal{D}$ is always ensured to be greater than or equal to one. Therefore, for each contract $p \in \mathcal{P}$, there exists a meta-action $d$ such that $p \in \mathcal{P}_d(\mathcal{D})$, thus ensuring that $\cup_{d \in \mathcal{D}} \mathcal{P}_d(\mathcal{D}) = \mathcal{P}$. This concludes the proof. $\qquad\square$

**Lemma 14.** *Under the event $\mathcal{E}_\epsilon$, it always holds $\mathcal{P}_d(\mathcal{D}) \subseteq \mathcal{U}_d$ for each meta-action $d \in \mathcal{D}$.*

*Proof.* To prove the lemma we observe that, for any meta-action $d_i \in \mathcal{D}$, the following inclusions hold:

$$\mathcal{U}_{d_i} = \cap_{j \in \mathcal{D}_{d_i}} \widetilde{H}_{ij} \supset \cap_{j \in \mathcal{D}_{d_i}} H_{ij} \supset \cap_{j \in \mathcal{D}} H_{ij} = \mathcal{P}_{d_i}(\mathcal{D}).$$

The first inclusion holds thanks to the definition of the halfspace $\widetilde{H}_{ij}$ and employing Lemma 10, which entails under the event $\mathcal{E}_\epsilon$. The second inclusion holds because $\mathcal{D}_{d_i}$ is a subset of $\mathcal{D}$ for each $d_i \in \mathcal{D}$. Finally, the last equality holds because of the definition of $\mathcal{P}_{d_i}(\mathcal{D})$. $\qquad\square$

**Lemma 5.** *Under the event $\mathcal{E}_\epsilon$, when Algorithm 3 returns $\{\mathcal{L}_d\}_{d \in \mathcal{D}}$, it holds that $\bigcup_{d \in \mathcal{D}} \mathcal{L}_d = \mathcal{P}$.*

*Proof.* As a first step we notice that, if Algorithm 3 returns $\{\mathcal{L}_d\}_{d\in\mathcal{D}}$, then the set $\mathcal{D}$ has not been updated during its execution and, thus, we must have $\mathcal{L}_d = \mathcal{U}_d$ for all $d \in \mathcal{D}$. In addition, we notice that, under the event $\mathcal{E}_\epsilon$, thanks to Lemma 13 and Lemma 14 the following inclusion holds:

$$\bigcup_{d_i\in\mathcal{D}} \mathcal{U}_{d_i} \supseteq \bigcup_{d_i\in\mathcal{D}} \mathcal{P}_{d_i}(\mathcal{D}) = \mathcal{P}.$$

Then, by putting all together we get:

$$\bigcup_{d_i\in\mathcal{D}} \mathcal{L}_{d_i} = \bigcup_{d_i\in\mathcal{D}} \mathcal{U}_{d_i} = \bigcup_{d_i\in\mathcal{D}} \mathcal{P}_i(\mathcal{D}) = \mathcal{P},$$

concluding the proof. $\qquad\square$

**Lemma 6.** *Under the event $\mathcal{E}_\epsilon$, when Algorithm 3 returns $\{\mathcal{L}_d\}_{d\in\mathcal{D}}$, for every meta-action $d \in \mathcal{D}$, contract $p \in \mathcal{L}_d$ and action $a' \in A(d)$, there exists a $\gamma$ that polynomially depends on $m$, $n$, $\epsilon$, and $B$ such that:*

$$\sum_{\omega\in\Omega} F_{a',\omega}\, p_\omega - c_{a'} \geq \sum_{\omega\in\Omega} F_{a,\omega}\, p_\omega - c_a - \gamma \quad \forall a \in \mathcal{A}.$$

*Proof.* In order to prove the lemma, we rely on the crucial observation that, for any vertex $p \in V(\mathcal{L}_{d_i})$ of the lower bound of a meta-action $d_i \in \mathcal{D}$, the `Action-Oracle` procedure called by Algorithm 3 with $p$ as input either returned $d_i$ or another meta-action $d_j \in \mathcal{D}$ such that $p \in \widetilde{H}_{ij}$ with $d_j \in \mathcal{D}_{d_i}$.

First, we consider the case in which the meta-action returned by `Action-Oracle` in the vertex $p$ is equal to $d_i$. In such a case, $a^\star(p) \in A(d_i)$ thanks to Lemma 1 and the following holds:

$$\sum_{\omega\in\Omega} \widetilde{F}_{d_i,\omega}\, p_\omega - c(d_i) \geq \sum_{\omega\in\Omega} F_{a^\star(p),\omega}\, p_\omega - c_{a^\star(p)} - 9B\epsilon mn,$$

by means of Lemma 4. Next, we consider the case in which the meta-action returned by Algorithm 2 is equal to $d_j$ with $p$ that belongs to $\widetilde{H}_{ij}$. In such a case the following inequalities hold:

$$\sum_{\omega\in\Omega} \widetilde{F}_{d_i,\omega}\, p_\omega - c(d_i) \geq \sum_{\omega\in\Omega} \widetilde{F}_{d_j,\omega}\, p_\omega - c(d_j) - y$$

$$\geq \sum_{\omega\in\Omega} F_{a^\star(p),\omega}\, p_\omega - c_{a^\star(p)} - 9B\epsilon mn - y,$$

where the first inequality follows by Lemma 10 that guarantees that $p$ belongs to $\widetilde{H}_{ij} \subseteq H_{ij}^y$ with $y = 18B\epsilon mn^2 + 2n\eta\sqrt{m}$, while the second inequality holds because of Lemma 4, since $a^\star(p) \in A(d_j)$ thanks to Lemma 1 .

Finally, by putting together the inequalities for the two cases considered above and employing Lemma 4, we can conclude that, for every vertex $p \in V(\mathcal{L}_d)$ of the lower bound of a meta-action $d \in \mathcal{D}$, it holds:

$$\sum_{\omega\in\Omega} F_{a,\omega}\, p_\omega - c_a \geq \sum_{\omega\in\Omega} \widetilde{F}_{d,\omega}\, p_\omega - c(d) - 9B\epsilon mn$$

$$\geq \sum_{\omega\in\Omega} F_{a^\star(p),\omega}\, p_\omega - c_{a^\star(p)} - \gamma, \tag{8}$$

for each action $a \in A(d)$ by setting $\gamma := 27B\epsilon mn^2 + 2n\eta\sqrt{m}$.

Moreover, by noticing that each lower bound $\mathcal{L}_d$ is a convex polytope, we can employ the Carathéodory's theorem to decompose each contract $p \in \mathcal{L}_d$ as a convex combination of the vertices of $\mathcal{L}_d$. Formally:

$$\sum_{p'\in V(\mathcal{L}_d)} \alpha(p')\, p'_\omega = p_\omega \quad \forall \omega \in \Omega, \tag{9}$$

where $\alpha(p') \geq 0$ is the weight given to vertex $p' \in V(\mathcal{L}_d)$, so that it holds $\sum_{p'\in V(\mathcal{L}_d)} \alpha(p') = 1$. Finally, for every $p \in \mathcal{L}_d$ and action $a \in A(d)$ we have:

$$\sum_{\omega\in\Omega} F_{a^\star(p),\omega}\, p_\omega - c_{a^\star(p)} = \sum_{\omega\in\Omega} F_{a^\star(p),\omega} \left( \sum_{p'\in V(\mathcal{L}_d)} \alpha(p')\, p'_\omega \right) - c_{a^\star(p)}$$

$$= \sum_{p' \in V(\mathcal{L}_d)} \alpha(p') \left( \sum_{\omega \in \Omega} F_{a^\star(p),\omega} \, p'_\omega - c_{a^\star(p)} \right)$$

$$\leq \sum_{p' \in V(\mathcal{L}_d)} \alpha(p') \left( \sum_{\omega \in \Omega} F_{a,\omega} \, p'_\omega - c_a + \gamma \right)$$

$$= \sum_{\omega \in \Omega} F_{a,\omega} \left( \sum_{p' \in V(\mathcal{L}_d)} \alpha(p') \, p'_\omega \right) - c_a + \gamma$$

$$= \sum_{\omega \in \Omega} F_{a,\omega} \, p_\omega - c_a + \gamma,$$

where the first and the last equalities hold thanks of Equation 9, while the second and the third equalities hold since $\sum_{p' \in V(\mathcal{L}_d)} \alpha(p') = 1$. Finally, the inequality holds thanks to Inequality 8 by setting $\gamma := 27 B \epsilon m n^2 + 2 n \eta \sqrt{m}$. $\qquad \square$

**Lemma 7.** *Under event $\mathcal{E}_\epsilon$, Algorithm 3 requires at most $\mathcal{O}\left(n^2 q \left(\log \left(Bm/\eta\right) + \binom{m+n+1}{m}\right)\right)$ rounds.*

*Proof.* As a first step, we observe that Algorithm 3 terminates in a finite number of rounds. By the way in which Algorithm 3 intersects the upper bounds with the halfspaces computed with the help of the Find-HS procedure, the algorithm terminates with $\mathcal{L}_d = \mathcal{U}_d$ for all $d \in \mathcal{D}$ after a finite number of rounds as long as, for each meta-action $d_i \in \mathcal{D}$, the halfspaces $\widetilde{H}_{ij}$ with $d_j \in \mathcal{D}$ are computed at most once. It is easy to see that this is indeed the case. Specifically, for every meta-actions $d_i \in \mathcal{D}$, Algorithm 6 is called to build the halfspace $\widetilde{H}_{ij}$ only when Algorithm 2 returns $d_j$ with $d_j \notin \mathcal{D}_{d_i}$ for a vertex $p \in V(\mathcal{U}_{d_i})$ of the upper bound $\mathcal{U}_{d_i}$. If the halfspace has already been computed, then $d_j \in \mathcal{D}_{d_i}$ by the way in which the set $\mathcal{D}_{d_i}$ is updated. As a result, if Algorithm 2 called on a vertex of the upper bound $\mathcal{U}_{d_i}$ returns the meta-action $d_j \in \mathcal{D}_{d_i}$, then Algorithm 3 does *not* compute the halfspace again.

For every $d_i \in \mathcal{D}$, the number of vertices of the upper bound $\mathcal{U}_{d_i}$ is at most $\binom{m+n+1}{m} = \mathcal{O}(m^n)$, since the halfspaces defining the polytope $\mathcal{U}_{d_i}$ are a subset of the $m+1$ halfspaces defining $\mathcal{P}$ and the halfspaces $\widetilde{H}_{ij}$ with $d_j \in \mathcal{D}$. The number of halfspaces $\widetilde{H}_{ij}$ is at most $n$ for every meta-action $d_i \in \mathcal{D}$. Consequently, since each vertex lies at the intersection of at most $m$ linearly independent hyperplanes, the total number of vertices of the upper bound $\mathcal{U}_{d_i}$ is bounded by $\binom{m+n+1}{m} = \mathcal{O}(m^n)$, for every $d_i \in \mathcal{D}$

We also observe that, for every $d_i \in \mathcal{D}$, the while loop in Line 28 terminates with at most $V(\mathcal{U}_{d_i}) = \binom{m+n+1}{m} = \mathcal{O}(m^n)$ iterations. This is because, during each iteration, either the algorithm finds a new halfspace or it exits from the loop. At each iteration of the loop, the algorithm invokes Algorithm 2, which requires $q$ rounds. Moreover, finding a new halfspace requires a number of rounds of the order of $\mathcal{O}\left(q \log \left(Bm/\eta\right)\right)$, as stated in Lemma 11. Therefore, the total number of rounds required by the execution of the while loop in Line 28 is at most $\mathcal{O}\left(q \left(\log \left(Bm/\eta\right) + \binom{m+n+1}{m}\right)\right)$.

Let us also notice that, during each iteration of the while loop in Line 9, either the algorithm finds a new halfspace or it exits from the loop. This is because, if no halfspace is computed, the algorithm does *not* update the boundaries of $\mathcal{U}_{d_i}$, meaning that the meta-action implemented in each vertex of $\mathcal{L}_{d_i}$ is either $d_i$ or some $d_j$ belonging to $\mathcal{D}_{d_i}$. Moreover, since the number of halfspaces $\widetilde{H}_{ij}$ is bounded by $n$ for each $\mathcal{U}_{d_i}$, the while loop in Line 9 terminates in at most $n$ steps. As a result, the number of rounds required by the execution of the while loop in Line 9 is of the order of $\mathcal{O}\left(nq \left(\log(Bm/\eta) + \binom{m+n+1}{m}\right)\right)$, being the while loop in Lines 28 nested within the one in Line 9.

Finally, we observe that the while loop in Line 7 iterates over the set of meta-actions actions $\mathcal{D}$, which has cardinality at most $n$. Therefore, the total number of rounds required to execute the entire algorithm is of the order of $\mathcal{O}\left(n^2 q \left(\log \left(Bm/\eta\right) + \binom{m+n+1}{m}\right)\right)$, which concludes the proof. $\qquad \square$

## C.4 PROOFS OF THE LEMMAS RELATED TO THE FIND-CONTRACT PROCEDURE

**Lemma 8.** *Under the event $\mathcal{E}_\epsilon$, if $\{\mathcal{L}_d\}_{d\in\mathcal{D}}$ is a cover of $\mathcal{P}$ computed by* Try-Cover*, Algorithm 4 returns a contract $p \in [0, B]^m$ such that $u(p) \geq \max_{p'\in[0,B]^m} u(p') - \rho$.*

*Proof.* In the following, we define $p^o \in [0, B]^m$ as the optimal contract, while we let $p^\ell := (1 - \sqrt{\gamma})p^o + \sqrt{\gamma}r$, where $\gamma$ is defined as in Lemma 6. Additionally, we define $d^o \in \mathcal{D}$ as one of the meta-actions such that $p^o \in \mathcal{L}_{d^o}$. Similarly, we let $d^\ell \in \mathcal{D}$ be one of the meta-actions such that $p^\ell \in \mathcal{L}_{d^\ell}$. It is important to note that $p^\ell \in [0, B]^m$ since $\|r\|_\infty \leq 1$. Furthermore, Lemma 5 ensures that there exists at least one $d^\ell \in \mathcal{D}$ such that $p^\ell \in \mathcal{L}_{d^\ell}$.

As a first step, we prove that, for each $a_i \in A(d^\ell)$, it holds:

$$
\begin{aligned}
\gamma &\geq \sum_{\omega\in\Omega}(F_{a^\star(p^\ell),\omega} - F_{a_i,\omega})p^\ell_\omega + c_{a_i} - c_{a^\star(p^\ell)} \\
&\geq \sum_{\omega\in\Omega}(F_{a^\star(p^o),\omega} - F_{a_i,\omega})p^\ell_\omega + c_{a_i} - c_{a^\star(p^o)} \\
&= \sum_{\omega\in\Omega}(F_{a^\star(p^o),\omega} - F_{a_i,\omega})p^o_\omega + c_{a_i} - c_{a^\star(p^o)} + \sqrt{\gamma}\sum_{\omega\in\Omega}(F_{a^\star(p^o),\omega} - F_{a_i,\omega})(r_\omega - p^o_\omega) \\
&\geq \sqrt{\gamma}\sum_{\omega\in\Omega}(F_{a^\star(p^o),\omega} - F_{a_i,\omega})(r_\omega - p^o_\omega),
\end{aligned}
$$

where the first inequality holds because of Lemma 6 since $p^\ell \in \mathcal{L}_{d^\ell}$, while the second and the third inequalities hold because of the definition of best-response and the equality holds because of the definition of $p^\ell \in \mathcal{L}_{d^\ell}$.

Then, by rearranging the latter inequality, we can show that for each action $a_i \in A(d^\ell)$ the following holds:

$$
\sum_{\omega\in\Omega} F_{a_i,\omega}(r_\omega - p^o_\omega) \geq \sum_{\omega\in\Omega} F_{a^\star(p^o),\omega}(r_\omega - p^o_\omega) - \sqrt{\gamma}. \tag{10}
$$

Furthermore, for each action $a_i \in A(d^\ell)$, we have that:

$$
\begin{aligned}
\sum_{\omega\in\Omega} F_{a_i,\omega}\left(r_\omega - p^\ell_\omega\right) &= \sum_{\omega\in\Omega} F_{a_i,\omega}\left(r_\omega - ((1-\sqrt{\gamma})p^o_\omega + \sqrt{\gamma}r_\omega)\right) \\
&= \sum_{\omega\in\Omega} F_{a_i,\omega}\left(r_\omega - p^o_\omega\right) - \sqrt{\gamma}\sum_{\omega\in\Omega} F_{a_i,\omega}\left(r_\omega - p^o_\omega\right) \\
&\geq \sum_{\omega\in\Omega} F_{a_i,\omega}\left(r_\omega - p^o_\omega\right) - \sqrt{\gamma} \\
&\geq \sum_{\omega\in\Omega} F_{a^\star(p^o),\omega}(r_\omega - p^o_\omega) - 2\sqrt{\gamma},
\end{aligned}
$$

where the first equality holds because of the definition of $p^\ell \in \mathcal{L}_{d_\ell}$, while the first inequality holds since $\|r\|_\infty \leq 1$ and the latter inequality because of Equation 10. Putting all together we get:

$$
\begin{aligned}
\sum_{\omega\in\Omega} F_{a_i,\omega}\left(r_\omega - p^\ell_\omega\right) &\geq \sum_{\omega\in\Omega} F_{a^\star(p^o),\omega}(r_\omega - p^o_\omega) - 2\sqrt{\gamma} \\
&= \mathsf{OPT} - 2\sqrt{\gamma}, \tag{11}
\end{aligned}
$$

for each $a_i \in A(d^\ell)$ with $p^\ell \in \mathcal{L}_{d^\ell}$.

Now, we show that the principal's utility in the contract returned by Algorithm 4 is close to the optimal one. To do that we let $\{\widetilde{F}_d\}_{d\in\mathcal{D}}$ be the set of empirical distributions employed by Algorithm 4. Furthermore, we let $p^\star \in \mathcal{L}_{d^\star}$ be the contract computed in Line 5 of Algorithm 4. Analogously, we let $p^f := (1 - \sqrt{\gamma})p^\star + \sqrt{\gamma}r$ be the final contract the principal commits to. Then, for each $a_i \in A(d^\star)$, we have:

$$
u(p^f) \geq \sum_{\omega\in\Omega} F_{a_i,\omega}\left(r_\omega - p^\star_\omega\right) - 2\sqrt{\gamma}
$$

$$\geq \sum_{\omega \in \Omega} \widetilde{F}_{d^\star, \omega} \left( r_\omega - p_\omega^\star \right) - 2\sqrt{\gamma} - 5\epsilon mn,$$

where the first inequality follows from Proposition A.4 by Dutting et al. (2021) and $u(p^\star) \leq 1$, while the second inequality holds by means of Definition 3 as $a_i \in A(d^\star)$. Analogously, for each $a_i \in A(d^\ell)$, we have:

$$\sum_{\omega \in \Omega} \widetilde{F}_{d^\star, \omega} \left( r_\omega - p_\omega^\star \right) \geq \sum_{\omega \in \Omega} \widetilde{F}_{d^\ell, \omega} \left( r_\omega - p_\omega^\ell \right)$$

$$\geq \sum_{\omega \in \Omega} F_{a_i, \omega} \left( r_\omega - p_\omega^\ell \right) - 5\epsilon mn$$

$$\geq \mathsf{OPT} - 2\sqrt{\gamma} - 5\epsilon mn,$$

where the first inequality holds because of the optimality of $p^\star$, the second inequality holds because of Definition 3 since $a_i \in A(d^\ell)$, while the third inequality holds because of Equation 11. Finally, by putting all together we get:

$$u(p^f) \geq \mathsf{OPT} - 4\sqrt{27 B \epsilon m n^2 + 2n\eta\sqrt{m}} - 10\epsilon mn$$

$$\geq \mathsf{OPT} - 32\sqrt{B\epsilon m^2 n^2},$$

where we employ the definition of $\gamma$ as prescribed by Lemma 6. As a result, in order to achieve a $\rho$-optimal solution we set:

$$\epsilon := \frac{\rho^2}{32^2 B m^2 n^2},$$

while $\eta := \epsilon \sqrt{m} n / 2$. $\qquad\qquad\qquad\qquad\qquad\qquad\qquad\qquad\qquad\qquad\qquad\qquad\qquad\quad\square$

## C.5 Proof of Theorem 2

**Theorem 2.** *Given $\rho \in (0, 1)$, $\delta \in (0, 1)$, and $B \geq 1$ as inputs, with probability at least $1 - \delta$ the* Discover-and-Cover *algorithm (Algorithm 1) is guaranteed to return a contract $p \in [0, B]^m$ such that $u(p) \geq \max_{p' \in [0, B]^m} u(p') - \rho$ in at most $\widetilde{\mathcal{O}}(m^n \cdot \mathcal{I} \cdot {}^1/_{\rho^4} \log({}^1/_\delta))$ rounds, where $\mathcal{I}$ is a term that depends polynomially in $m$, $n$, and $B$.*

*Proof.* First, we notice that to achieve $\rho$-optimal solution under the event $\mathcal{E}_\epsilon$, as observed in Lemma 8, we must set:

$$\epsilon := \frac{\rho^2}{32^2 B m^2 n^2} \quad \text{and} \quad \eta := \epsilon \sqrt{m} n / 2. \qquad (12)$$

To ensure that Algorithm 1 returns a $\rho$-optimal solution with a probability of at least $1 - \delta$, we need to set the remaining parameters $\alpha$ and $q$ in a way that $\mathbb{P}(\mathcal{E}_\epsilon) \geq 1 - \delta$. Intuitively, the probability of the event $\mathcal{E}_\epsilon$ corresponds to the probability that, whenever Algorithm 2 is invoked by Algorithm 3, it returns an empirical distribution sufficiently close to the actual one.

First, we observe that given $\epsilon, \alpha$ and a distribution over outcomes $F$, Algorithm 2 computes an empirical distribution $\widetilde{F}$ satisfying $\|\widetilde{F} - F\|_\infty \leq \epsilon$ with a probability of at least $1 - \alpha$, in a number of rounds $q = \lceil {}^1/_{2\epsilon^2} \log \left( {}^{2m}/_\alpha \right) \rceil$ as prescribed by Lemma 12.

To ensure that Algorithm 2 returns an empirical distribution that closely approximates the true distribution each time it is called, we need to bound the number of times the Discover-and-Cover procedure invokes Algorithm 2. By applying Lemma 7, we have that the maximum number of times the Action-Oracle algorithm is called by the Try-Cover algorithm is bounded by $n^2 \left( \log \left( {}^{2Bm}/_\eta \right) + \binom{m+n+1}{m} \right)$. Additionally, according to Lemma 2, the Try-Cover procedure is invoked at most $2n$ times during the execution of the Discover-and-Cover algorithm. Consequently, the number of times Algorithm 2 is invoked is bounded by $2n^3 \left( \log \left( {}^{2Bm}/_\eta \right) + \binom{m+n+1}{m} \right)$.

By applying a union bound over all the times Algorithm 2 is invoked and considering that each time it returns an empirical distribution that is within $\epsilon$ distance in the $\| \cdot \|_\infty$ norm from the actual distribution with probability at least $1 - \alpha$, we can conclude that the event $\mathcal{E}_\epsilon$ occurs with probability at least:

$$\mathbb{P}(\mathcal{E}_\epsilon) \geq 1 - 2\alpha n^3 \left( \log \left( \frac{2Bm}{\eta} \right) + \binom{m+n+1}{m} \right).$$

As a result, by setting:

$$\alpha := \frac{\delta}{2n^3 \left( \log \left( 2Bm/\eta \right) + \binom{m+n+1}{m} \right)},$$ (13)

with $\eta$ defined as above guarantees that $\mathbb{P}(\mathcal{E}_\epsilon) \geq 1 - \delta$. Thus, the number of rounds $q$ required by Algorithm 2 is equal to:

$$q := \left\lceil \frac{1}{2\epsilon^2} \log \left( \frac{2m}{\alpha} \right) \right\rceil,$$

with $\epsilon, \alpha$ defined as in Equations 12 and 13. Then, by employing Lemma 2 and Lemma 7, the number of rounds to execute Algorithm 1 is of the order of $\mathcal{O} \left( qn^3 \left( \log \left( 2Bm/\eta \right) + \binom{m+n+1}{m} \right) \right)$.

Finally, by definition of the parameters $\alpha, \epsilon$, and $q$, the total number of rounds required by Algorithm 1 to return a $\rho$-optimal solution with probability at least $1 - \delta$ is at most:

$$\widetilde{\mathcal{O}} \left( m^n \frac{B^2 m^4 n^8}{\rho^4} \log \left( \frac{1}{\delta} \right) \right),$$

which concludes the proof. $\qquad\square$

## D   OTHER OMITTED PROOFS

In this section, we provide all the remaining omitted proofs.

**Theorem 1.** *For any number of rounds $N \in \mathbb{N}$, there is no algorithm that is guaranteed to find a $\kappa$-optimal contract with probability greater than or equal to $1 - \delta$ by using less than $N$ rounds, where $\kappa, \delta > 0$ are some suitable absolute constants.*

*Proof.* We consider a group of instances parametrized by a parameter $\epsilon \in (0, 1/80)$. In each instance, we let $\mathcal{A} = \{a_1, a_2\}$ be the set of actions while we let $\Omega = \{\omega_1, \omega_2, \omega_3\}$ be the set of outcomes. Furthermore, the distributions over the outcomes of the two actions are defined as follows: $F_{a_1} = (1/2, 0, 1/2)$ and $F_{a_2} = (0, \epsilon, 1 - \epsilon)$ with associated cost of $c_{a_1} = 0$ and $c_{a_2} = 1/4$, respectively. In all the instances the principal's reward is given by $r = (0, 0, 1)$ while the optimal contract is equal to $p^* = (0, 1/4\epsilon, 0)$, resulting in a principal's expected utility of $u(p^*) = 3/4 - \epsilon$.

As a first step, we show that if $p_{\omega_2} \leq 1/8\epsilon$, then the principal's utility is at most $9/80$-optimal. To show that, we first consider the case in which the agent selects action $a_1$. In such a case, the highest expected utility achieved by the principal is at most $1/2$, which occurs when they commit to the null contract $p = (0, 0, 0)$. Clearly, the utility achieved in $p = (0, 0, 0)$ is not $9/80$-optimal, for each possible $\epsilon \in (0, 1/80)$.

Then, we consider the case in which the agent selects action $a_2$ and $p_{\omega_2} \leq 1/8\epsilon$. In this scenario, the highest principal's expected utility is attained in the contract $p' = (0, 1/8\epsilon, 1/(4-8\epsilon))$. Thus, the resulting principal's expected utility is at most:

$$u(p') = \sum_{\omega \in \Omega} F_{a_2, \omega}(r_\omega - p'_\omega) \leq -\frac{1}{8} + (1 - \epsilon) \left( 1 - \frac{1}{4 - 8\epsilon} \right) < -\frac{1}{8} + \frac{3}{4} = \frac{5}{8},$$

which is not $9/80$-optimal, for any value of $\epsilon \in (0, 1/80)$. Consequently, for each possible action selected by the agent, if $p_{\omega_2} \leq 1/8\epsilon$, then the expected utility of the principal's utility cannot be $9/80$-optimal.

To conclude the proof, we consider two instances characterized by $\epsilon_1 = 1/(80N \log(2N))$ and $\epsilon_2 = 1/(80N^2)$, for an arbitrary fixed $N \geq 1$. In the following, we let $\mathbb{P}_{\epsilon_1}$ and $\mathbb{P}_{\epsilon_2}$ be the probability measures induced by the N-rounds interconnection of an arbitrary algorithm executed in the first and in the second instances, respectively. Furthermore, we denote with $\mathcal{KL}(\mathbb{P}_{\epsilon_1}, \mathbb{P}_{\epsilon_2})$ the Kullback-Leibler divergence between these two measures. Then, by applying the Kullback-Leibler decomposition, with a simple calculation we can show that:

$$\mathcal{KL}(\mathbb{P}_{\epsilon_1}, \mathbb{P}_{\epsilon_2}) \leq \mathbb{E}_{\epsilon_1} \left[ \sum_{t=1}^{N} \mathcal{KL}(F_{a_2}^{\epsilon_1}, F_{a_2}^{\epsilon_2}) \right]$$

$$\leq N(\epsilon_1 \log(\epsilon_1/\epsilon_2) + (1 - \epsilon_1) \log((1-\epsilon_1)/(1-\epsilon_2)))$$
$$\leq 2/79,$$

where we let $F_{a_i}^{\epsilon_1}$ and $F_{a_i}^{\epsilon_2}$ be the distributions over outcomes of action $a_i \in \mathcal{A}$ in the first and in the second instances, respectively.

We now introduce the event $\mathcal{I}$, defined as the event in which the final contract returned by a given algorithm satisfies the condition $p_{\omega_2} \geq 1/8\epsilon_2$. We observe that if the event $\mathcal{I}$ holds in the first instance, then the learned contract provides a negative principal's utility. On the contrary, if such an event does not hold in the second instance, the final contract is not $9/80$-optimal, as previously observed. Then, by the Pinsker's inequality we have that:

$$\mathbb{P}_{\epsilon_2}(\mathcal{I}^c) + \mathbb{P}_{\epsilon_1}(\mathcal{I}) \geq \frac{1}{2} \exp\left(-\mathcal{KL}(\mathbb{P}_{\epsilon_1}, \mathbb{P}_{\epsilon_2})\right) = \frac{1}{2} \exp\left(-2/79\right). \tag{14}$$

Consequently, there exists no algorithm returning a $9/80$-optimal with a probability greater or equal to $1/4 \exp(-2/79)$, thus concluding the proof. $\square$

**Theorem 3.** *Given $\alpha \in (0, 1)$, Algorithm 5 achieves $R^T \leq \widetilde{\mathcal{O}}\left(m^n \cdot \mathcal{I} \cdot \log(1/\delta) \cdot T^{4/5}\right)$ with probability at least $1 - \delta$, where $\mathcal{I}$ is a term that depends polynomially on $m$, $n$, and $B$.*

*Proof.* Thanks to Theorem 2, we know that by employing an appropriate number of rounds, the solution returned by Algorithm 1 is $\rho$-optimal with probability at least $1 - \delta$, for given values of $\rho$ and $\delta$ greater than zero. Furthermore, we notice that the per-round regret suffered by Algorithm 5 is bounded by $mB + 1$ during the execution of Algorithm 1, and it is at most $\rho$ for the remaining rounds. Formally, we have that:

$$R_T \leq \widetilde{\mathcal{O}}\left(m^n \frac{B^3 m^5 n^8}{\rho^4} \log\left(\frac{1}{\delta}\right) + T\rho\right).$$

Thus, by setting $\rho = m^{n/5} B^{3/5} mn^{8/5} T^{-1/5}$ as input to Algorithm 1, with probability at least $1 - \delta$ the cumulative regret is bounded by:

$$R_T \leq \widetilde{\mathcal{O}}\left(m^n B^{3/5} n^{8/5} \log\left(\frac{1}{\delta}\right) T^{4/5}\right),$$

concluding the proof. $\square$

