# OpenReview forum: "Learning Optimal Contracts: How to Exploit Small Action Spaces"
_ICLR.cc/2024/Conference — ICLR 2024 poster_

### Official Review · Reviewer_NM4o · 2023-10-28

**Soundness:** 3 good
**Presentation:** 3 good
**Contribution:** 2 fair
**Rating:** 6
**Confidence:** 3

**Summary:**

This paper study the sample complexity (and its implication in online learning) of learning the optimal contract in the principal-agent problem. Specifically, they consider the setting where the principal, who does not know the forecast matrix of the agent, offers the agent a contract, and the agent performs the action that maximizes the agent's own utility, and then the principal observes the outcome as a result of the action performed by the agent but not the action itself. The principal may choose different contracts and repeat the above process to figure out the optimal non-negative and bounded contract that maximizes the principal's utility, and the number of repetitions is the sample complexity of this problem.

The main result of this paper is an algorithm that has $\tilde{O}(m^n)$ sample complexity, where $m$ is the number of the outcomes, and $n$ is the number of the actions of the agent. In particular, the sample complexity is polynomial in $m$ when $n$ is a constant.

The key observation underlying this result is that the polytope $\mathcal{P}_i$ containing exactly all the contracts to which the best response of the agent is the same action $i\in[n]$, is defined by $m+n$ inequalities ($m+1$ inequalities that force the contract to be non-negative and bounded, and $n - 1$ inequalities that force the action $i$ to be better than other actions for the agent). Hence, $\mathcal{P}_i$ has at most $\binom{m+n}{m}=\binom{m+n}{n}\le (m+n)^n$ vertices.

The algorithm starts with an outer approximation $\mathcal{P}_i'$ of $\mathcal{P}_i$ ($\mathcal{P}_i'$ is initially defined by the $m+1$ inequalities that force the contract to be non-negative and bounded), and as long as there is some vertex $v$ of $\mathcal{P}_i'$ that is not in $\mathcal{P}_i$ (in other words, the agent's best response to the contract $v$ is some $j\neq i$), it can easily find an inequality (using the structure of the principal-agent problem) that approximately corresponds to the constraint ``action $i$ is better than action $j$ for the agent'', and then it adds this inequality to $\mathcal{P}_i'$. After the algorithm iteratively finds $n-1$ such inequalities, $\mathcal{P}_i'$ will be approximately same as $\mathcal{P}_i$. Moreover, to find one such constraint for the current iterate $\mathcal{P}_i'$, the algorithm only needs to evaluate at most all the contracts corresponding to all the vertices of $\mathcal{P}_i'$, the number of which is bounded $(m+n)^n$ by the key observation above.

**Strengths:**

* The result is original, and the idea is simple and nice.
* The writing is good overall.

**Weaknesses:**

I think the biggest weakness of this paper is that it studies the setting without prior distribution of the agent's type.
* First of all, I feel that the sample complexity problem is not well-motivated in this setting -- If I'm the principal, when an agent comes, I won't try to find my optimal contract merely for this agent by signing the agent multiple times with different contracts and observing the outcomes. I would only use this approach, when there is a large candidate pool, and I do not know the prior distribution of the candidates' skills.
* Moreover, the previous work of [Zhu et al. 2022] studies the setting with prior distribution of the agent's type. Their open question is also posed in that setting, so I think ``solves an open problem by Zhu et al.'' is an overclaim.

On a separate note, the space of contract considered in this paper is different from that in [Zhu et al. 2022]. In this paper, the contract is bounded in the sense that the sum of the payments for all outcomes is bounded by some number $B$, but in [Zhu et al. 2022], the contract is bounded in the sense that the payment of each outcome is bounded by some number $B$. That is, even in the narrow setting without prior distribution, the problem studied by this paper is not quite the same as the original one.

**Questions:**

Could you address the comments in the weakness section?

---

> ### Author Response · Authors · 2023-11-15
>
> >First of all, I feel that the sample complexity problem is not well-motivated in this setting -- If I'm the principal, when an agent comes, I won't try to find my optimal contract merely for this agent by signing the agent multiple times with different contracts and observing the outcomes. I would only use this approach, when there is a large candidate pool, and I do not know the prior distribution of the candidates' skills.
>
> We disagree with the Reviewer on this point, as we think that the sample complexity problem that we study in our paper is well motivated. Indeed, we believe that, even in the case where the principal repeatedly interacts with the same agent, if the principal has no prior knowledge about agent's features, then it is reasonable that the principal would try to learn them by repeatedly signing contracts with the agent. Honestly, we are not able to find a better approach to tackle the problem. At the same time, the problem is arguably interesting and of practical relevance even with a single type.
>
> Moreover, let us also remark that similar learning problems have also been addressed for the (much simpler) Stackelberg problems, where the leader repeatedly interacts with the same (unknown) follower in order to learn an optimal strategy to commit to (Letchford et al. (2009); Peng et al. (2019)). Followers do not have types in these settings as well.
>
> Another way of looking at the sample complexity problem studied in our paper is to imagine a scenario in which the principal has access to some ``simulated model'' of the agent that can be used to learn agent's features and in turn an optimal contract to commit to when the actual interaction with the agent takes place. Thus, we believe that studying learning in principal-agent problems in which the agent has a single type is of interest and well motivated.
>
> > Moreover, the previous work of [Zhu et al. 2022] studies the setting with prior distribution of the agent's type. Their open question is also posed in that setting, so I think ``solves an open problem by Zhu et al.'' is an overclaim.
>
> We agree with the Reviewer. In the final version of the paper, we will make explicit that our work provides an answer to the open question posed by Zhu et al. (2022) in the specific case in which there is a single agent's types.  Nonetheless, the lower bound by Zhu et al. (2022) holds for the setting with a single type, proving that the setting with arbitrary number of actions is intractable even in single-type instances. This suggests that at least some of the main difficulties of the problem are present in instances with a single type, motivating our study. We believe that our result is a first milestone towards answering the open question by Zhu et al. (2022) in more general multi-type settings. Indeed, when seeking for solutions to open problems, it is reasonable to make the first attempts in settings that are more specific than those in which the question was originally posed.
>
> > On a separate note, the space of contract considered in this paper is different from that in [Zhu et al. 2022]. In this paper, the contract is bounded in the sense that the sum of the payments for all outcomes is bounded by some number B, but in [Zhu et al. 2022], the contract is bounded in the sense that the payment of each outcome is bounded by some number B. That is, even in the narrow setting without prior distribution, the problem studied by this paper is not quite the same as the original one.
>
> Let us remark that, in our paper, the problem of learning an approximately-optimal contract is framed for the contract space defined by the hypercube $[0,B]^m$ with $B \geq 1$, which is strictly more general that the contract space $[0,1]^m$ considered by Zhu et al. (2022). Thus, our algorithm is guaranteed to return a contract in $[0,B]^m$ (see the Find-Contract sub-procedure). To do so, our algorithm defines other sets of contracts. In particular, we exploit the set of contracts whose $1$-norm is bounded. This is a clever trick to reduce the number of hyperplanes defining the polytope of contracts. We remark that the returned contract lies in $[0,B]^m$ as in Zhu et al. (2022).

---

> > ### Comment · Reviewer_NM4o · 2023-11-22
> >
> > Thanks for clarifying the last point about the contract space.

---

> > > ### Author Response · Authors · 2023-11-22
> > >
> > > Thanks for your response. We hope that we have also addressed the other Reviewer's points and that the Reviewer is willing to reconsider the evaluation of our paper.

---

### Official Review · Reviewer_GCN2 · 2023-10-31

**Soundness:** 3 good
**Presentation:** 3 good
**Contribution:** 2 fair
**Rating:** 6
**Confidence:** 3

**Summary:**

This paper studies the principal-agent problem where a principal seeks to induce an agent to take a beneficial but unobservable action through contracts. They aim to learn an optimal contract by observing outcomes. The paper proposes an algorithm that can learn an approximately optimal contract within a polynomial number of rounds related to the outcome space size. Specifically, they showed an algorithm with a sample complexity result of O(m^n T^4/5) and also converted it to an explore-then-commit online learning algorithm with sublinear regret.

**Strengths:**

Learning the optimal principal’s strategy in principal-agent problems when the agent’s type is unknown has become an important problem in those kinds of games with sequential movements. Existing works mainly focused on Stackelberg (security) games. This paper introduced meta-actions to group together the agent’s actions associated with “similar” distributions over outcomes for the specific contract design problem where the agent’s action is unobservable. Then this paper demonstrates how to utilize this idea to learn the approximately optimal contract. The results seem well-executed and the characterization is interesting.

**Weaknesses:**

One major modeling concern I have is the assumption that the agent will honestly best respond to the principal’s queries, especially if they know that the principal is learning to play against him. This problem is particularly an issue in contract design — if the principal does not know the agent’s utility and wants to learn to play against the agent, then the agent would have strong incentives to manipulate their responses to mislead the principal to learn some non-optimal contracts and would like to do so to benefit himself. Could you elaborate a bit about the validity of the honest agent best response assumption, and some application domains where this assumption holds?

On the technical side, I feel like the contribution of this paper might be limited given Peng et al., [2019] and Zhu et al., [2022]. “The core idea of the algorithm is to progressively build two polytopes Ud and Ld” is highly similar to the algorithm proposed by Peng et al. [2019], except that here the agent’s action is not observable, so the authors replace it with some meta action, which is realized by the Action-Oracle and does not seem to be enough contribution for ICLR. As for the relaxed assumption regarding the constant volume, which seems to be highly dependent on the result Lemma 4 by Zhu et al. [2022], which shows the Continuity of the principal’s utility function.

**Questions:**

Could you provide some application domains where your honest-responding agent behavior holds?


How is your Lemma 8 different from Lemma 4 by Zhu et al. [2022]? I feel like your results can be directly implied from their result about the Continuity of the principal’s utility function.

---

> ### Author Response · Authors · 2023-11-15
>
> > One major modeling concern I have is the assumption that the agent will honestly best respond to the principal’s queries, especially if they know that the principal is learning to play against him. This problem is particularly an issue in contract design — if the principal does not know the agent’s utility and wants to learn to play against the agent, then the agent would have strong incentives to manipulate their responses to mislead the principal to learn some non-optimal contracts and would like to do so to benefit himself. Could you elaborate a bit about the validity of the honest agent best response assumption, and some application domains where this assumption holds?
>
> First, let us remark that the assumption that the agent plays a best response after observing the principal's commitment is a standard assumption in the literature on principal-agent problems, even in online settings. See, as an example, the works by Ho et al. (2014); Dutting et al. (2019, 2021, 2022); Alon et al. (2021); Guruganesh et al. (2021);  Castiglioni et al. (2022a;b; 2023); Zhu et al. (2022). We are not aware of any work in which the agent does not play a best response, even if the are plenty of works on online settings (see, e.g., Ho et al. (2014), Cohen et al. (2022), Zhu et al. (2022)). Moreover, such an assumption is made also in the related literature on sample complexity in Stackelberg games (see, e.g., Letchford et al. (2009); Peng et al. (2019)).
>
> We agree with the Reviewer that a very smart agent can manipulate their responses. However, this would assume an unreasonably strong knowledge of the principal by the agent and a strong assumption on the behavior of the principal. Similar problems have been studied in simpler settings under strong assumptions (see, e.g., Deng et al. (2019)). Indeed, if the agent lacks knowledge about the principal's reward function and misreports their true best response, then the final contract that the principal commits to may yield lower utility to the agent, compared to the case in which they truthfully respond. Consequently, the agent has no reason to misreport their actual best response if they have no knowledge about the principal's reward function.
>
> To summarize, we agree with the Reviewer that it would be interesting and realistic to study settings in which the agent manipulates the principal. However, the current advancement of the research towards this direction is not mature enough and this problem is unexplored even in much simpler settings.
>
> [Deng, Yuan, Jon Schneider, and Balasubramanian Sivan. ``Strategizing against no-regret learners." Advances in neural information processing systems 32 (2019)]
>
>
> > Could you provide some application domains where your honest-responding agent behavior holds?
>
> We remark that this assumption is common in all works on repeated principal-agent problems (see, e.g., Ho et al. (2014), Cohen et al. (2022), Zhu et al. (2022)). Indeed, the assumption that the agent plays a best response after observing the principal's commitment is also made in several works that study applications of principal-agent problems. For instance, Ho et al. (2014) study their application to crowdsourcing platforms, while Bastani et al. (2016) study a principal-agent problem with an application in healthcare settings.
>
> > How is your Lemma 8 different from Lemma 4 by Zhu et al. [2022]? I feel like your results can be directly implied from their result about the Continuity of the principal’s utility function.
>
>  Lemma 4 by Zhu et al. (2022) is related to the Lipschitz continuity of the principal's utility function along some directions. This result is not applicable in our framework. Indeed, our Lemma 8 relates the principal's utility obtained under approximately-incentive-compatible contracts and incentive-compatible ones. Furthermore, this is only a small component of our proof and not the main result. Indeed, the main contribution is to show that the principal's utility obtained by "undiscovered" action is not large with respect to those of "discovered" actions.

---

> ### Author Response · Authors · 2023-11-15
>
> > On the technical side, I feel like the contribution of this paper might be limited given Peng et al., [2019] and Zhu et al., [2022]. “The core idea of the algorithm is to progressively build two polytopes Ud and Ld” is highly similar to the algorithm proposed by Peng et al. [2019], except that here the agent’s action is not observable, so the authors replace it with some meta action, which is realized by the Action-Oracle and does not seem to be enough contribution for ICLR. ...
>
>
> In the following, we summarize the technical and algorithmic challenges arising in our setting with respect to the one studied by Letchford et al. (2009) and Peng et al. (2019). These challenges mainly arise from the fact that, differently from the Stackelberg case in which one can directly observe the agent's best-response action at the end of each round, in our setting the principal only observes an outcome that is stochastically determined (according to an unknown distribution) as an effect of such an action. This introduce two additional difficulties: we cannot compute the exact separating hyperplanes defining best-response regions and we cannot identify the true set of actions. These difficulties would make the problem impossible to solve in Stackelberg games (e.g., it could be the case that some actions are never discovered). However, we can overcome these difficulties in principal-agent problems by using the fact that the principal's utility enjoys some particular structures that are not present in Stackelberg games. As a result, a direct application of the algorithmic approach by Peng et al. (2019) would fail in our setting. While our approach builds on the same idea of building lower and upper bounds for the agent's best response regions, on a technical perspective our approach is quite different and more involved (as evidence, notice that our solution includes several algorithms and pages of proofs while the algorithm of Peng et al. (2019) can be described and analyzed in a few pages).
>
>
>    In the following, we summarize the technical challenges in extending the results of Letchford et al. (2009) and Peng et al. (2019):
>
>
> - The Action-Oracle procedure implements several checks that allow to associate the observed empirical distributions to existing meta-actions, discover new meta-actions, or merge existing ones. In particular, the Action-Oracle procedure must adapt and redefine the set of meta-actions to ensure that each meta-action consistently includes agent's actions having a similar distribution over outcomes (see Lemma 1) and limit the number of invocations to Try-Cover (see Lemma 2). We observe that Lemma 1 requires a non-trivial technical effort in order to ensure that the Action-Oracle procedure does not merge empirical distributions that form a “chain” growing arbitrarily in terms of infinity-norm.
>
> - The Try-Cover procedure requires much effort compared to the the procedure employed by Peng et al. (2019). For instance, the Try-Cover algorithm involves several additional checks to ensure that the procedure terminates appropriately when Action-Oracle updates the set of meta-actions, and the meta-action implemented in the vertex of the lower bound belongs to a suitably-defined set of previously-computed hyperplanes. The need for these additional checks arises from the fact that, in our setting with unobservable actions, the approach proposed by Peng et al. (2019) would not terminate within a finite number of steps. As a result, it becomes challenging to prove that: the final collection of lower bounds is a coverage of the entire space of contracts (Lemma 5), the meta-actions at the vertexes of the final lower bounds are effectively approximate best responses (Lemma 6), and the algorithm terminates in a finite number of steps (Lemma 7).
>
> - The Try-Cover algorithm relies on a suitably-defined procedure to determine the separating hyperplane between two meta actions (see Find-HS in Appendix B). Such a procedure is made effective thanks to Lemma 9 and Lemma 10. These ensure that the intercepts of the computed hyperplanes are close to the differences between the costs of the two corresponding meta-actions, as defined in Definition 4.
>
> - In our algorithm, the final lower bounds do not coincide with the actual best-response regions. Consequently, the guarantee that an optimal commitment lies within one of the vertices of the final lower bounds, as it is the case in Peng et al. (2019), is not assured. To address this problem, we introduce two non-trivial lemmas (Lemma 6 and 8) showing that the utility of an optimal contract is close to the one returned by the Discover-and-Cover algorithm. These lemmas require non-trivial effort and rely on the specific properties that are guaranteed by both the Action-Oracle and the Try-Cover procedures. See Appendix C.

---

> > ### Comment · Reviewer_GCN2 · 2023-11-22
> >
> > I appreciate the detailed response from the authors. For the issue of what constitutes sufficient contribution I agree, this is in the eye of the beholder and I am willing to concede this point. I raised my score to a marginal accept.

---

### Official Review · Reviewer_Mt7g · 2023-11-01

**Soundness:** 3 good
**Presentation:** 3 good
**Contribution:** 3 good
**Rating:** 6
**Confidence:** 4

**Summary:**

This paper studies the problem of learning optimal contracts in a principal-agent setting with focus on the setup where the size of the agent's action space is small.

**Strengths:**

- This paper improves the previous sample complexity bounds for learning optimal contracts in the case when the number of actions is constant.

**Weaknesses:**

- It seems to me that the approach of this paper seems to be mostly based on the work of [Letchford et al., 2009; Peng et al., 2019], but this is not explicitly mentioned in the paper. This is acceptable to me, but I expect the authors to discuss and summarize the challenges of extending the previous approach to the current setting.

some minor comments:

- Theorem 1 should rather be an observation or proposition than a theorem, since it is a quite obvious fact in learning optimal contracts, even though it might be mentioned explicitly in previous work.

- Definition 1 should rather be an assumption than a definition. I also do not see any benefit of treating this condition as a high probability event. I think you are essentially restricting the learning problem to a subclass of instances where the approximately optimal contract has bounded payment.

- The current notation is not very readable to me, and I would suggest the authors make some efforts on simplifying them. For example, you could use dot product instead of element-wise product over the outcomes spaces in many places.

**Questions:**

- Do we know any lower bounds for the sample complexity of learning optimal contracts when the action size is small? Do you think it is possible to improve the current method beyond the constant action size case (possibly under other assumptions)?

---

> ### Author Response · Authors · 2023-11-15
>
> > It seems to me that the approach of this paper seems to be mostly based on the work of [Letchford et al., 2009; Peng et al., 2019], but this is not explicitly mentioned in the paper. This is acceptable to me, but I expect the authors to discuss and summarize the challenges of extending the previous approach to the current setting.
>
>
> In the following, we summarize the technical and the algorithmic challenges arising in our setting with respect to the one studied by Letchford et al. (2009) and Peng et al. (2019). These challenges mainly arise from the fact that, differently from the Stackelberg case in which one can directly observe the agent's best-response action at the end of each round, in our setting the principal only observes an outcome that is stochastically determined (according to an unknown distribution) as an effect of such an action. This introduce two additional difficulties: we cannot compute the exact separating hyperplanes defining best-response regions and we cannot identify the true set of actions. These difficulties would make the problem impossible to solve in Stackelberg games (e.g., it could be the case that some actions are never discovered). However, we can overcome these difficulties in principal-agent problems by using the fact that the principal's utility enjoys some particular structures that are not present in Stackelberg games. As a result, a direct application of the algorithmic approach by Peng et al. (2019) would fail in our setting. While our approach builds on the same idea of building lower and upper bounds for the agent's best-response regions, on a technical perspective our approach is quite different and more involved (as evidence, notice that our solution includes several algorithms and pages of proofs while the algorithm of Peng et al. (2019) can be described and analyzed in a few pages).
>
>
> In the following, we summarize the technical challenges in extending the results of Letchford et al. (2009) and Peng et al. (2019):
>
> - The Action-Oracle procedure implements several checks that allow to associate the observed empirical distributions to existing meta-actions, discover new meta-actions, or merge existing ones. In particular, the Action-Oracle procedure must adapt and redefine the set of meta-actions to ensure that each meta-action consistently includes agent's actions having a similar distributions over outcomes (see Lemma 1) and also limit the number of invocations of Try-Cover (see Lemma 2). We observe that Lemma 1 requires a non-trivial technical effort in order to ensure that the Action-Oracle procedure does not merge empirical distributions that form a “chain” growing arbitrarily in terms of infinity-norm.
>
> - The Try-Cover procedure requires much more effort compared to the procedure employed by Peng et al. (2019). For instance, the Try-Cover algorithm involves several additional checks to ensure that the procedure terminates appropriately when Action-Oracle updates the set of meta-actions, and the meta-action implemented in the vertex of the lower bound belongs to a suitably-defined set of previously-computed hyperplanes. The need for these additional checks arises from the fact that, in our setting with unobservable actions, the approach proposed by Peng et al. (2019) would not terminate within a finite number of steps. As a result, it becomes challenging to prove that: the final collection of lower bounds is a coverage of the entire space of contracts (Lemma 5), the meta-actions found in the vertexes of the final lower bounds are effectively approximate best responses (Lemma 6), and the algorithm terminates in a finite number of steps (Lemma 7).
>
> - The Try-Cover algorithm relies on a suitably-defined procedure to determine the separating hyperplane between two meta-actions (see Find-HS in Appendix B). Such a procedure is made effective thanks to Lemma 9 and Lemma 10. These ensure that the intercepts of the computed hyperplanes are close to the differences between the costs of the two corresponding meta actions, as defined in Definition 4.
>
> - In our algorithm, the final lower bounds do not coincide with the actual best-response regions. Consequently, the guarantee that an optimal commitment lies within one of the vertices of the final lower bounds, as it is the case in Peng et al. (2019), is not assured. To address this problem, we introduce two non-trivial lemmas (Lemma 6 and 8) showing that the utility of an optimal contract is close to the one returned by the Discover-and-Cover algorithm. These lemmas require non-trivial effort and rely on the specific properties that are guaranteed by both the Action-Oracle and the Try-Cover procedures. See Appendix C.

---

> ### Author Response · Authors · 2023-11-15
>
> > Theorem 1 should rather be an observation or proposition than a theorem, since it is a quite obvious fact in learning optimal contracts, even though it might be mentioned explicitly in previous work.
>
> We agree with the Reviewer that the result in Theorem 1 is somehow expected. Nevertheless, to the best of our knowledge, no previous works have formally proved it. Moreover, the proof is not so trivial to be done formally. That said, we agree with the Reviewer that turning Theorem 1 into a proposition is the right choice.
>
> > Definition 1 should rather be an assumption than a definition. I also do not see any benefit of treating this condition as a high probability event. I think you are essentially restricting the learning problem to a subclass of instances where the approximately optimal contract has bounded payment.
>
> We believe that the Reviewer is not correctly interpreting Definition 1. Indeed, the definition formalizes the problem solved in our paper, which is the one of learning an approximately-optimal bounded contract, adopting a PAC-learning-inspired perspective. We remark that this problem is meaningful in any instance (also those in which the optimal contract sets payments above the bound).
>  Moreover, notice that the `with-high-probability' requirement in Definition 1 is necessary in order to meaningfully define the learning task. This is different from what we believe the Reviewer is considering, i.e., assuming to restrict the attention to problem instances where there exists an approximately-optimal contract with bounded payments. As a result, we think that Definition 1 should remain a definition (and not turning into an assumption). Moreover, let us also remark that our problem (Definition 1) is consistent with what has been studied in previous works (see, e.g., the work by Zhu et al. (2022)).
>
> > The current notation is not very readable to me, and I would suggest the authors make some efforts on simplifying them. For example, you could use dot product instead of element-wise product over the outcomes spaces in many places.
>
> We thank the Reviewer for the observation. We will take it into consideration in the final version of the paper.
>
> > Do we know any lower bounds for the sample complexity of learning optimal contracts when the action size is small? Do you think it is possible to improve the current method beyond the constant action size case (possibly under other assumptions)?
>
> When the number of actions is constant, our algorithm provides a bound that is polynomial in the instance size. We think that this is to be considered as satisfactory and tight (meaning that there exists a trivial lower bound polynomial in the instance size). It is known that without additional assumptions the sample complexity is exponential (Zhu et al., 2022).
>  Nevertheless, we believe that this lower bound can be circumvented under some assumptions.
>  For instance, assuming that the best-response regions either have a finite constant volume or are empty, we conjecture that it is possible to design an algorithm that depends on the smallest of such volumes and do not require an exponential dependence on the number of agent's actions.

---

### Meta-Review · Area_Chair_Dxkb · 2023-12-04

**Metareview:**

This paper considers a online variant of the classical contract problem between a principal and an agent.

I believe the main contribution is the conceptual one, not necessarily the technical ones (even though they are not trivial). I think learning optimal contract - in an online fashion here -, is highly relevant for other communities close to the one of ICLR (thinking of EC for instance).  One might argue that ICLR is not the optimal venue for such a paper, but I would disagree. It might provide new directions to some participants, and I strongly feel that those directions are worth investigating.

The reviewers all agree to give the same score of 6, marginally above acceptance, hence the poster recommendation.

**Justification For Why Not Higher Score:**

Interesting, maybe a bit niche for ICLR

**Justification For Why Not Lower Score:**

N/A

---

### Decision · Program_Chairs · 2024-01-16

Accept (poster)